# Reducing societal impacts of SARS-CoV-2 interventions through subnational implementation

Mark M Dekker[1,2,3]\*, Luc E Coffeng[4], Frank P Pijpers[5,6], Debabrata Panja[1,2†], Sake J de Vlas[4†]

[1]Department of Information and Computing Sciences, Utrecht University, Utrecht, Netherlands; [2]Centre for Complex Systems Studies, Utrecht University, Utrecht, Netherlands; [3]PBL Netherlands Environmental Assessment Agency, The Hague, Netherlands; [4]Department of Public Health, Erasmus MC, University Medical Center Rotterdam, Rotterdam, Netherlands; [5]Statistics Netherlands, The Hague, Netherlands; [6]Korteweg-de Vries Institute for Mathematics, University of Amsterdam, Amsterdam, Netherlands

\*For correspondence:
m.m.dekker@uu.nl

†These authors contributed equally to this work

Competing interest: The authors declare that no competing interests exist.

**Abstract** To curb the initial spread of SARS-CoV-2, many countries relied on nation-wide implementation of non-pharmaceutical intervention measures, resulting in substantial socio-economic impacts. Potentially, subnational implementations might have had less of a societal impact, but comparable epidemiological impact. Here, using the first COVID-19 wave in the Netherlands as a case in point, we address this issue by developing a high-resolution analysis framework that uses a demographically stratified population and a spatially explicit, dynamic, individual contact-pattern based epidemiology, calibrated to hospital admissions data and mobility trends extracted from mobile phone signals and Google. We demonstrate how a subnational approach could achieve similar level of epidemiological control in terms of hospital admissions, while some parts of the country could stay open for a longer period. Our framework is exportable to other countries and settings, and may be used to develop policies on subnational approach as a better strategic choice for controlling future epidemics.

## Editor's evaluation

This useful study, based on simulations from an agent-based model of SARS-CoV-2 transmission in the first wave in the Netherlands, provides solid evidence that a subnational implementation of non-pharmaceutical interventions, with control measures varying in response to local variations in infection prevalence, would have the potential to achieve similar levels of epidemiological control to a nationally homogeneous response while reducing negative societal impacts. The work will be of interest to communicable disease epidemiologists and those involved in policy responses to epidemics.

## Introduction

As in many countries around the world (*Brauner et al., 2021*; *Liu et al., 2021*), control of the first COVID-19 pandemic wave in the Netherlands was largely based on nation-wide implementation of a variety of non-pharmaceutical intervention measures (e.g. lockdown, social distancing, or reduced mobility). Their associated societal burden affected all areas in the country equally, while infections and the healthcare burden, in contrast, were distributed heterogeneously across space and time. This

brings in focus the question whether the pandemic could have been controlled equally well with interventions specifically tailored to subnational regions, such as municipalities or provinces. In addition to preventing the unnecessary broader societal burden of interventions in (largely unaffected) parts of a country, such tailored strategies potentially have several additional advantages: (1) more efficient use of resources, such as test kits and mobile laboratories; (2) reduced economic losses due to interventions; (3) reducing intervention-adherence fatigue in the population.

Epidemiological analyses can help to explore the value of such strategies (*Walker et al., 2020*). However, the challenge therein lies in the fact that epidemiological dynamics cannot easily be untangled from human behavior, which varies strongly across societies and cultures (*Walker et al., 2020*), and are highly heterogeneous even within a population living in a certain geographic region (*Marks et al., 2021*; *Davies et al., 2020*). For this reason, such an epidemiological analysis not only needs to capture the spatio-temporal heterogeneities in both transmission and control of an infectious disease, but also 'to embed itself locally' (*Vos et al., 2020*; *Eggo et al., 2021*): the demographic composition of the population and how people travel, interact and mingle, across different demographic groups and subnational regions (*Lloyd-Smith et al., 2005*; *Prem et al., 2017*; *Mossong et al., 2008*). Building a corresponding analysis framework that takes all this into account is however not only highly complex, but also requires rich data at high resolutions.

Such challenges have left their vivid marks in the first COVID-19 wave. By and large, intervention measures deployed in spring 2020 were not enough to spatially contain the virus: the worldwide spread of SARS-CoV-2 along the backbones of globalized travel was too fast to allow continuation of travel as usual. Reliable data (specifically, near-real time data needed for policy-informing epidemiology) on community-transmission were not readily available to researchers and policy makers during most part of the first wave. For setting intervention policies in such a situation, large parts of the world used epidemiological insights that were emerging from other countries that experienced the epidemic earlier, notably China (*Li et al., 2020*; *Zhang et al., 2020*). First, this meant that local embedding was being missed (*Eggo et al., 2021*). Second, by the time reliable data started to become available, national policies in many countries, for example, the Netherlands (*RIVM, 2021a*) or the UK (*Kucharski et al., 2020*), were mostly informed by models considering populations that were demographically but not spatio-temporally heterogeneous (*Press and Levin, 2020*).

Here, using the Netherlands as a case in point, and supported by a combination of rich data sources (demography, mobility, mixing, hospitalization and seroprevalence), we develop an epidemiological analysis of the first COVID-19 wave by building a dynamic proxy network of people's contacts to embed into the local context as well as to account for high-resolution spatio-temporal heterogeneities (*Cevik and Baral, 2021*). The wave covers the period February 27, 2020 (the first tested case of COVID-19 in the Netherlands) till June 1, 2020 (lifting of most intervention measures). In relation to the celebration of Carnaval, an annual festivity preceding Lent that is heavily celebrated in the south of the country and is associated with large group gatherings and movement of people, the outbreak started mainly in the south of the Netherlands. In this timeline, there are four distinguishable periods in terms of the policy landscape, which we refer to as *phases*: (i) Phase 1 (Feb 27 - Mar 11) when transmission of the pathogen progressed unchecked, (ii) Phase 2 (Mar 12 - Mar 22) with minor interventions involving a working-from-home policy, cancellation of large events, some social distancing and face mask advice in specific buildings such as hospitals, (iii) Phase 3 (Mar 23 - May 11) involving a strict nation-wide lockdown with closed schools and event centers, mandated social distancing and working-from-home policies, and (iv) Phase 4 (May 11 - May 31) involving a gradual lifting of all measures. The analysis not only allows us to individually assess the efficacy of the (national) non-pharmaceutical intervention measures that were implemented in the Netherlands, but it also allows us to investigate to what extent subnational implementation of interventions during the first wave of COVID-19 would have led to poorer or comparable control of the pandemic in the country as a whole. In larger countries the most appropriate subnational resolution could be at the level of counties, provinces, or any other existing administrative regions to make best use of clear lines of communication and responsibilities; in a small, densely populated country like the Netherlands, municipalities are the most appropriate ones. Our analysis can be exported to any other country provided comparably rich datasets, capturing the local embedding for the analysis, are available.

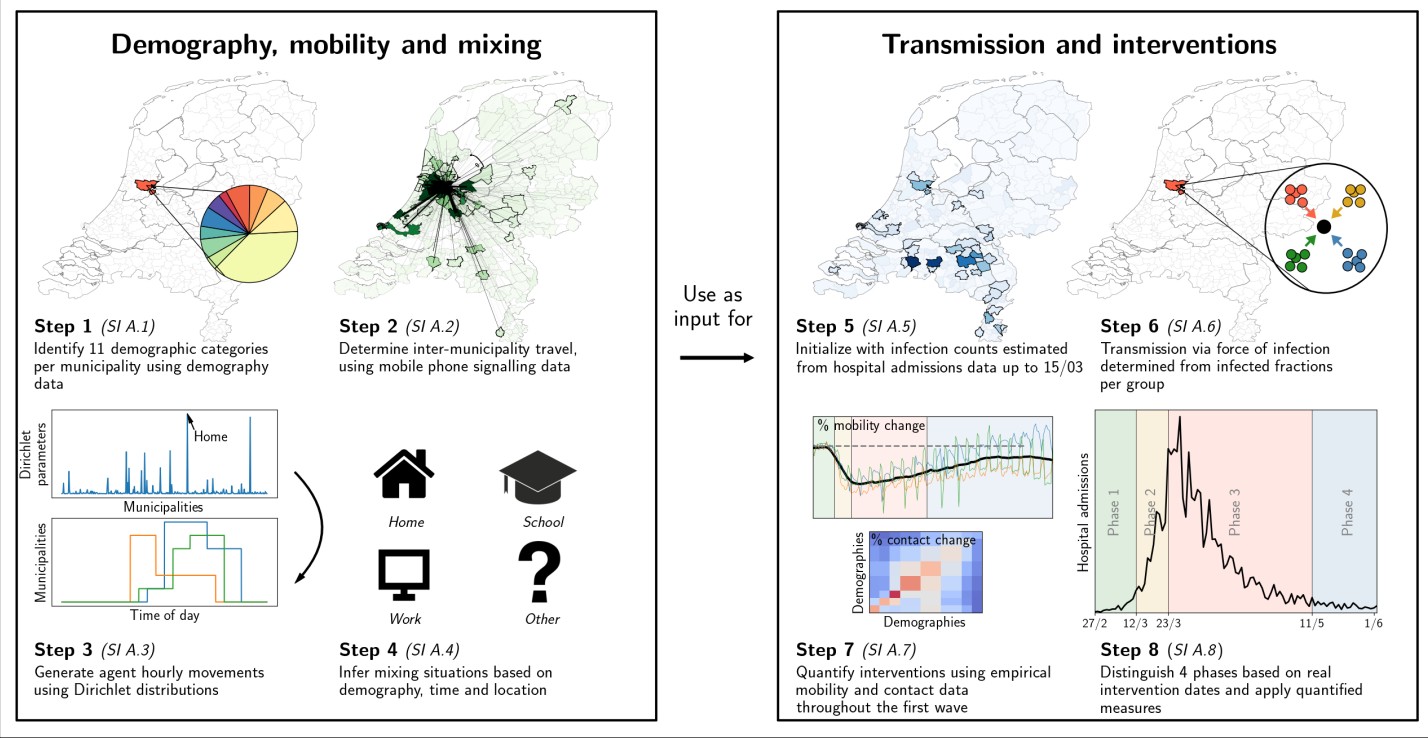

**Figure 1.** Our analysis framework consists of two parts: establishing proxy dynamic contact patterns from information on demography, mobility and mixing (left panel), and transmission and interventions (right panel); each part consists of four further steps. See Appendix 2.1 for a description of the data used in steps 2 and 5. Processes in steps 2, 3, and 6 are stochastic in nature.

## Results

### Analysis framework

Taking an agent-based approach, we build our framework in two parts: (i) demography, mobility and mixing considerations that provide a high-degree of local embedding, and (ii) transmission and interventions, each consisting of four steps (1–4 and 5–8, respectively in *Figure 1*). The key steps for the epidemiological dynamics are summarized below; additional details can be found in the methods section and Appendix 1.1-1.8 (one Appendix 1 section per step).

In the first part, we define the agents and their movements. In the first step, using registry data available at the Dutch national statistics agency (CBS, Statistics Netherlands), we stratify the Dutch population into 11 demographic categories and 380 municipalities. With about 17 million Dutch residents, we define an *agent* to represent approximately 100 Dutch residents. We distribute the agents across municipalities proportionally to population sizes. The second step is to define the probability that an agent moves between municipalities. This process is constructed using Dirichlet distributions for the probability of an agent's location, quantified based on anonymized mobile phone signals. In the third step, we draw the agent's locations and movements at hourly time resolution. The fourth step is to define the mixing of agents present within the same municipality, which depends on the demographic category of the agent, time of day and the type of activity that the agent is engaged in: 'home', 'school', 'work', and 'other'. The corresponding mixing matrices were based on existing surveys (*Prem et al., 2017*). Together, the four steps establish a dynamic proxy network of people's contacts throughout the entire country at municipality-level, with hourly resolution over the full period of analysis.

The second part of the analysis covers transmission and interventions. Here, the fifth step concerns the initialization of the epidemic transmission model, which was based on observed hospital admissions, which initially occurred mainly in the south of the country. The sixth step was to define transmission, based on the SEIR model for agent-to-agent pathogen transmission, which means that every agent at any time has one of the following four labels: susceptible ($S$), exposed ($E$), infectious ($I$) and recovered ($R$). Every 1-hr time step, susceptible agents may move to the exposed compartment as a

result of the *force of infection* that they experience as a function of the prevalence of infectious cases in each demographic category in the same municipality, expected contact rates between the agent and the different demographic categories, and their respective infectiousness. The seventh step concerns the quantification of changes across the first COVID-19 wave: (i) behavioral measures that reduce contact rates, (ii) mobility reductions, and (iii) school closure. Mobility changes were computed using Google Mobility data and mixing changes were based on survey data (*Backer et al., 2021*) conducted during this period. The effect of behavioral measures were calibrated to reproduce the epidemic trend over time. In the final step, we simulate transmission and the effect of changes in interventions over time. Predicted trends in infection numbers were translated to incident and prevalent hospital admission using a simple cohort model (*de Vlas and Coffeng, 2021*) that accounts for the delay between initial infection and admission as well as the duration of admission. This cohort model was quantified based on hospitalization data from the Dutch National Intensive Care Evaluation (NICE) registration. (Henceforth, at any point of time, we refer to individuals that have been exposed in the past as 'affected', so that at that point in time, they are either exposed, infectious, or recovered.)

A summary of the analysis itself can be found in the Methods section.

**Figure 2.** Calibration (**a–b**), and demography- and geography-resolved results from our analysis (**c–e**). Panel (**a**), left axis: the daily number of new infections and exposures in yellow and green, respectively. Right axis: daily hospital admissions from analysis output (red) and observed data (black). Background colors and vertical black lines denote the four phases (arbitrary coloring). Uncertainty intervals mark the minima and the maxima in the ensemble of realizations used in the analysis; the same holds for panels (**b**), (**c**) and (**e**). Panel (**b**): Hospitalization doubling time over the period March 13 - March 27, 2020 (shaded gray shaded time domain) in analysis (red, 4.6 days) and observed data (black, 4.61 days). Panel (**c**): % affected agents (i.e., $E$, $I$ or $R$) per demographic group for March 12 (dashed) and March 23 (solid). Panel (**d**): % affected agents per municipality on two days (March 5, May 25). Blue circles indicate the geographical locations of the three example municipalities shown in panel (**e**). Panel (**e**): Infectious agents (yellow) and hospital-admitted agents (analysis in red, and observed data in black) in three municipalities: Eindhoven, The Hague and Groningen. Analysis data correspond to an ensemble of 10 independent realizations.

## Reproducing the first COVID-19 wave

Even for a geographically heterogeneous analysis it is necessary to verify that the national trends are reproduced, which serves to calibrate and validate the relevant parameters in our simulations. The results of the calibration process, carried out by means of an ensemble of 40 stochastic simulations, is shown *Figure 2(a–b)*. The calibration is performed by means of four transmission-related parameters — $\beta_1$ through $\beta_4$, one for each phase of the first wave — to reproduce the *total national* hospital admissions data spanning approximately three months [panel (a)], including the (initial) doubling time [panel (b)]. Hospital admissions were the most reliable source of data during the first wave, and are shown in *Figure 2* as a thick black line in both panels, with the red line and its margins showing the range produced by our simulations. The curves in other colors in panel (a) denote the numbers of infectious and exposed people, obtained from simulations. See Materials and methods for the $\beta_t$-parameter values, and Appendix 1.8 for the details of the calibration process.

Age stratification in our analysis reveals how the first wave likely played out nationally across demographic groups, with non-studying adolescents, middle-age working people, and students as the most affected demographic groups [*Figure 2(c)*]. The model predicts similar patterns for seroprevalence levels across age as was observed in June 2020 [*Appendix 2—figure 1*]. It also predicts that the epidemic geographically spread from the south (where COVID-19 is introduced in the analysis) to the north of the country via major cities in the west [*Figure 2(d)*]. This geographic pattern approximately reflects the actual spread in the Netherlands, although we should not expect the analysis to perfectly reproduce given the high variability in the ensemble runs (see Appendix 2.4). Finally, in panel 2(e) the hospitalization data over time are compared for three different locations in the Netherlands: the first Dutch outbreak site in the South (Eindhoven), a location in the West (The Hague) where the epidemic spread relatively quickly, and a site in the North (Groningen) which was affected less and also later. That only four (national-level $\beta_t$-) parameters leads to realistic geographical spread across 380 individual municipalities over time serves to validate our approach for a geographically heterogeneous analysis (next section).

After the satisfactory calibration process above, we use the analysis to unravel the impact of individual lockdown components (behavior, mobility, school closure). *Figure 3(a)*, again a 40-member strong ensemble, shows how reductions in mobility contributed most to epidemic control; without mobility restrictions (red), case numbers would have approximately doubled. Behavioral changes (blue) have also had a considerable impact, albeit lower than mobility. (Determining the impact of the behavioral intervention measures is fairly straightforward: rather than varying the values of the transmission-related parameters $\beta_1$-$\beta_4$, we simply keep all at the same value as for the very first phase.) Our analysis also predicts school closure [yellow, *Figure 3(a)*] to have had little impact. On this, we note that due to political debate, the Dutch schools were closed relatively late (March 16, while the first confirmed

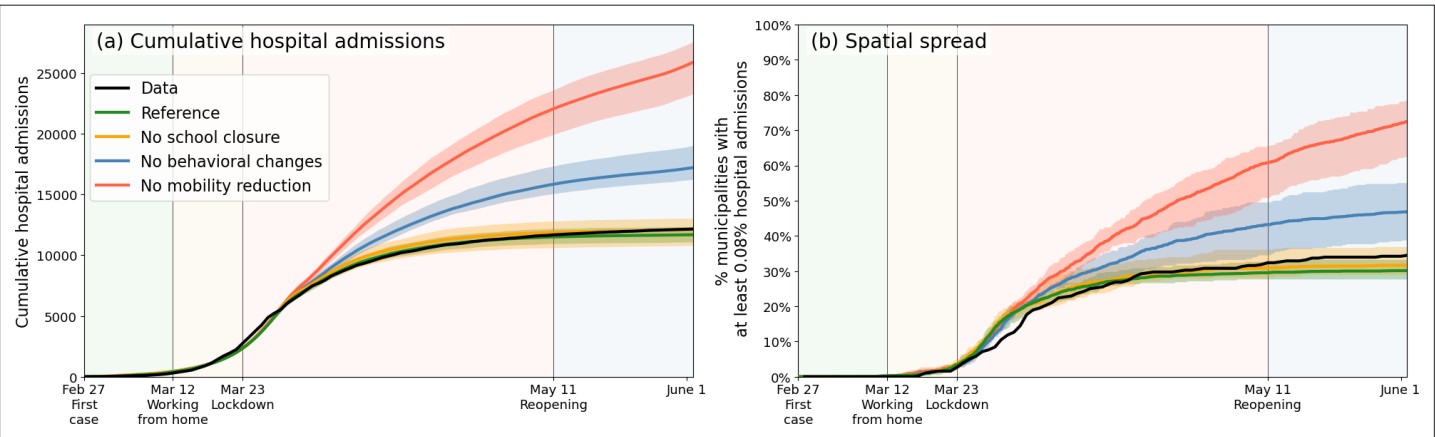

**Figure 3.** Comparing the impacts of nationally administered intervention measures. In both panels, observed data are shown in black, the reference in green, and the impacts of (i) no behavioral changes like wearing masks, enhanced hygiene and social distancing in blue, (ii) no mobility reduction in red and (iii) no closing of schools in yellow. Bandwidths indicate the minima and maxima around the mean of a simulation ensemble of 40 realizations. Panel (**a**): Cumulative national hospital admissions. Panel (**b**): Geographical spread of hospital admissions, measured by the fraction of municipalities that have at least 0.08% of the population admitted to the hospital.

case was on Feb 27) and therefore have contributed little to epidemic control in our analysis (logically, earlier closure of schools should have had a positive epidemiological impact, see Appendix 2.3). The individual lockdown components contributed similarly to spatial spread [*Figure 3(b)*], which quantifies the geographic spread of the COVID-19 pandemic in the Netherlands by following the number of municipalities affected substantially (for this, we use the measure of having > 0.08% of population hospital admitted).

## Effects of subnational implementation of interventions

Next, we evaluate the potential of subnational interventions, which in the Dutch case concerns non-pharmaceutical interventions issued at the level of municipalities. For a fair comparison across scenarios and with hospital admission data during the first wave, we implement subnational interventions in our simulations following the national trend. This means that we initiate lockdown in a municipality when the simulated prevalence of infectious cases within that municipality has passed a certain threshold — a fraction of the municipality's population — where the exact intervention measures are synchronous those issued in reality on a national scale (Appendix 1.9). Choosing the value of this threshold poses a trade-off: a lower threshold ensures implementation of local interventions in an early stage of the COVID-19 wave which would suppress hospital admission counts, but could unnecessarily shut down economic and social activity in some parts of the country that are less affected by the disease. Vice versa, a higher threshold would target municipalities where the epidemic has progressed most, but could pose the risk of starting control too late, resulting in more hospital admissions. To show the effect of different thresholds for prevalence of infectious cases, we choose a wide range of 3%, 1%, 0.33%, and 0.1%. Our choice to use prevalence of infectious cases for local decision-making is motivated by the following premise. Even though testing and case reporting were not yet at a sufficient scale to inform local decisions during the first wave, since then they were significantly scaled up. Moreover, with emerging methods and technologies such as sewage monitoring, fast identification of disease biology (e.g. time until symptoms) and live tracking of infections by mass testing (*Pavelka et al., 2021*) and using apps, number of infections in future will be proxy-estimated with progressively greater accuracy and speed, facilitating faster decision-making on subnational intervention measures (such as, in the Netherlands, starting or scaling down lockdowns at the level of municipalities).

The results are shown in *Figure 4*. In panel (a), the epidemiological impact of subnational interventions is quantified in terms of the number of hospital admissions, while the societal impact is quantified in panel (b) by the number of municipalities that are undergoing interventions. In panel (a), the lockdown as implemented in the Netherlands is represented by the black (observed) and green lines (prediction), which resulted in approximately 13 thousand hospital admissions up to 1 June 2020. Higher thresholds for deciding to implement a local lockdown clearly result in higher numbers of cumulative hospital admissions [panel (a)] and correspondingly a lower number of municipalities affected [panel (b)], and vice versa. A decision-making threshold of 3% (dark red) can be seen to be too high; although it only selects a few municipalities to go into lockdown directly at March 12th (185 million additional person-days intervention-free over the full wave), which could be considered a benefit of this approach, it results in a 157% increase in number of admissions ($\sim$ 19 thousand). The more stringent thresholds of 1.0% and 0.33% result in numbers of hospital admissions closer to a national lockdown (4670 and 410 additional admissions, respectively), but at a more modest societal benefit: 268 and 167 municipalities initialize interventions later than in the national approach, respectively. This translates to 103 million and 36 million additional person-days free from interventions over the full period. Interestingly, at the lower threshold of 0.33% (orange), approximately 6% of the municipalities never undergo interventions. Even closer to the fully national approach, we also tested a threshold of 0.1% (yellow), which yields only a few additional hospital admissions, and still 18 municipalities remaining intervention-free for over 5 weeks. The maps [*Figure 4(c–d)*] show the corresponding geographical distribution of percentages of affected people [panel (c)] and the societal benefits of subnational interventions in terms of the fraction of simulation-ensemble realizations in which a municipality remains without interventions [panel (d)]. Municipalities that remain free of interventions are mainly located in the north and east of the country, as can be most clearly seen for the 0.33% threshold scenario. From a mobility perspective, these municipalities belong to the more rural, isolated, and less densely populated subnational regions of the country.

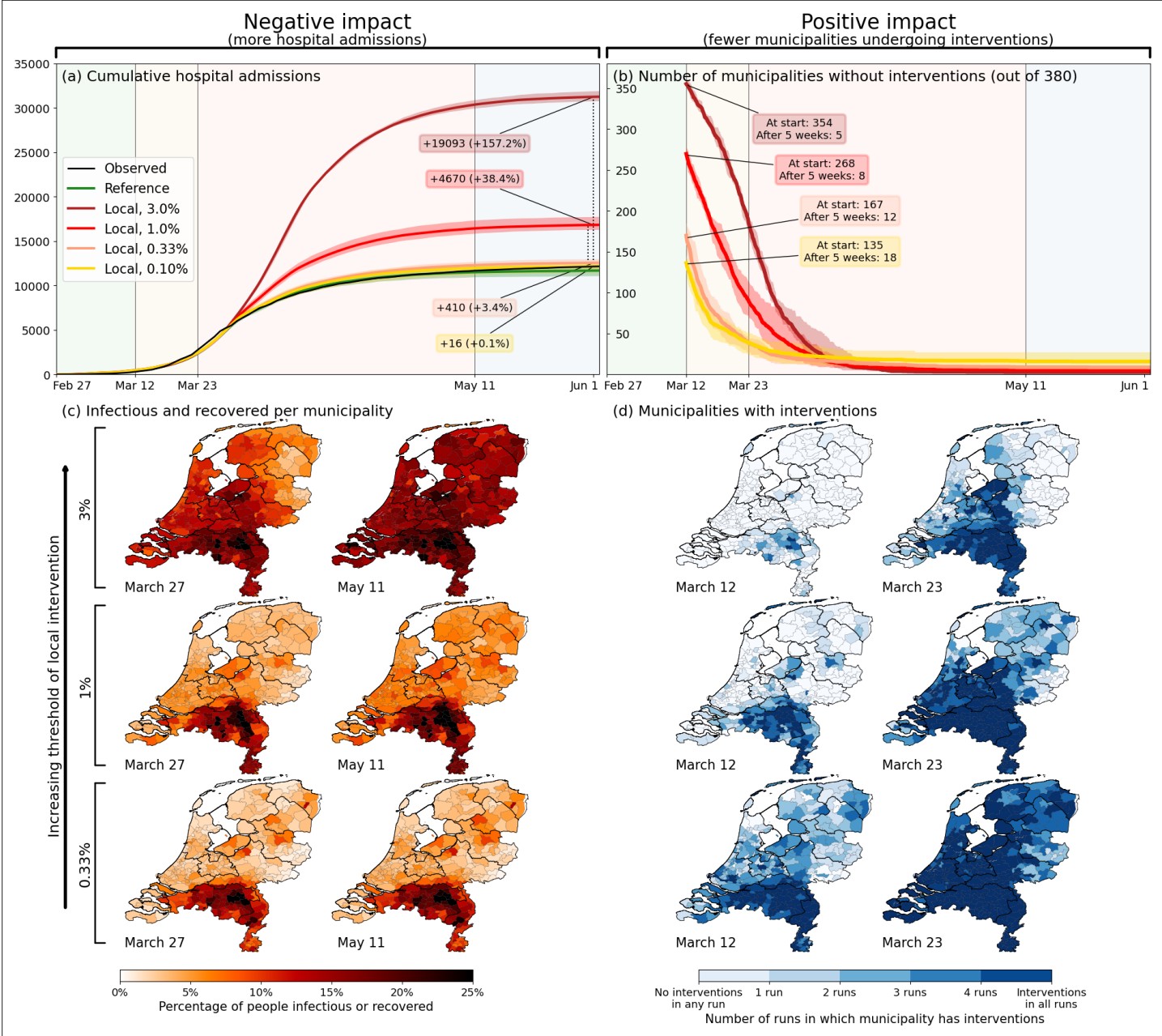

**Figure 4.** Quantification of the trade-off between costs (left) and benefits (right) of locally-adjusted interventions at four threshold values (0.1%, 0.33%, 1% and 3% of population simultaneously infectious). Panel (**a**): Cumulative hospital admissions for different scenarios. Panel (**b**): Fraction of municipalities that do not have any interventions in place. Panel (**c**): Cumulative fraction of infection cases per municipality for the three local intervention thresholds. The additional number and percentage growth in hospital admissions as compared to the observed national interventions is indicated. Panel (**d**): Geographical indication of which municipalities have undergone interventions and which ones not. The number of municipalities that do not is shown in panel (**b**).

During the first COVID-19 wave, the Dutch government did not implement subnational intervention measures, aside from bringing out an early advice to work from home in the south of the country, the epicenter for the first wave. The reasoning was that once COVID-19 cases were discovered locally, most likely, the pathogen would have already spread throughout the entire country. This is generally in line with observations that the Netherlands is spatially well-connected in terms of people's mobility patterns, facilitated by a robust public transport system and a high population density, with the caveat that hospital admission data during the first wave did suggest that provinces in the north and east of the country were substantially less affected. Our results show that when combined with live tracking

of local infections in sufficient detail, implementation of interventions could be postponed or tailored towards local contexts, without causing too much additional health burden. (This would of course require local governments to be mandated appropriately and that local populations adhere to local measures.)

## Discussion

Using the Netherlands as a case in point, we have evaluated the contribution of different interventions to the total effect of the lockdown, and explored to what extent subnational implementations of intervention measures might have had less of a societal impact, but comparable epidemiological impact. To this end, we have developed a highly detailed geographically- and demographically stratified analysis framework based on a dynamic proxy network of people's contacts throughout the entire country at municipality-level, with hourly resolution, which in turn utilizes human mobility between municipalities based on mobile phone signal data. We found that in the Netherlands, mobility reductions during the first wave contributed most to epidemic control; without them, we predict that a doubling of hospital admissions would have occurred. Our analysis, albeit based on a small country, shows that subnational (translated to be at the municipality-level) implementation of interventions strategies is worth considering, provided that means to monitor infection levels are available (via sewage surveillance [*Mallapaty, 2020*]), can substantially reduce the societal burden of interventions. The benefits of such an approach are expected to be even greater for larger and more populated countries. Moreover, similar or even higher gains can be expected by considering a subnational approach for also lifting interventions at a subnational level: analogous to initializing interventions, the reduction of the disease's prevalence across municipalities is not synchronous and, depending on the chosen prevalence threshold, some will be able to lift earlier than was done nationally.

Even though the methodology proposed in this paper comprises demographic and geographic stratification, and distinguishes multiple circumstances of mixing, there are still forms of granularity that we omit (e.g. households), which limits our ability to evaluate the impact of specific interventions with higher precision (*Lee et al., 2020*). For instance, when incorporating the effect of school closures, the effect of interacting only with family members instead of schoolmates has been captured at the level of a municipality as a whole (i.e. a different mixing pattern between demographic groups combined with an overall lower transmission rate). As such, our framework cannot provide insights into the role of households and household-level interventions, which have for instance been shown to play a critical role in the geographical spread of infection between schools (*Munday et al., 2021*; *Lessler et al., 2021*). Another limitation is that mobility in our framework is quantified based on mobile phone signal data that only provide anonymized movements between pairs of locations. As such, the data do not provide identifiers to link multiple movements into one itinerary, which means that in our analysis, agent movements are somewhat shorter on average than in reality, but agents also visit more different locations than in reality. We further assume that agent movements vary randomly day-by-day, whereas in reality commuting means that an agent would repeatedly travel to the same location. However, the impact of this simplifying assumption is limited as, at the start of an epidemic, the distribution of movement over agents is of relatively low importance, especially in the case of a relatively small and highly connected country as the Netherlands. This is in contrast to situations towards the tail of an epidemic or in larger geographies (e.g. Brazil [*Castro et al., 2021*; *de Souza et al., 2020*] and India [*Laxminarayan et al., 2020*]), where the transmission potential of 'high-mobility corridors' can eventually dry up as a result of rising immunity among high-mobility individuals. Finally, we adopted data on national patterns in mobility (Google mobility), meaning that it was not possible to account for changes in mobility by geography or demographic group. The geographical aspects could be addressed by using longitudinal mobile phone signal data or individual-level self-reported data via mobile phone apps (*Drew et al., 2020*; *Oliver et al., 2020*; *Ferretti et al., 2020*). This would require that such data are stored in a useful and accessible format in a General Data Protection Regulation (GDPR)-compliant manner, which may be challenging indeed.

In this study, we investigated only one of the several potential uses of our framework in a specific country. With appropriate data sources, the framework can be adapted to other countries and settings of similar or larger geographical scale. Importantly, the framework can also address other policy questions that involve a geographical or social dimension. For instance, we explored the potential impact of specifically isolating affected subnational areas (i.e. banning all mobility into and out of a municipality

for the Netherlands), which could reduce hospital admissions by about 30%, compared to the actual national lockdown (Appendix 2.3). With further expansions, the framework could address questions related to, for instance, closing or limiting specific (public) transport routes (*Quilty et al., 2020*) and banning specific mass events (*Herng et al., 2022*; *Moritz et al., 2021*; *Lemieux et al., 2021*) — for both of which much more fine-grained (temporal and geographical) data would be required. Evaluating pharmaceutical interventions such as vaccination, too, is possible to capture within this framework, upon coupling data sources associated with age-stratified vaccination rollout, as well as types of vaccines used.

In conclusion, we have demonstrated the potential added value of subnational implementation of interventions which, with appropriate information about infection levels in subnational areas, may significantly reduce the societal burden of lockdowns to control infectious disease. For the Dutch case, we calculate explicitly how many municipalities could have remained open with limited additional hospital admissions: 167 at the start and 12 still open after five weeks, with only 3.4% more hospital admissions. Of course, these numbers cannot be directly projected in subsequent COVID-19 waves in the Netherlands, or for that matter, to any waves in other countries or other variants and diseases. Nevertheless, there are several merits of this study for a broader context to mention. First, the policy relevance of our study is that we highlight the potential of subnational interventions. The Netherlands has a high population density and is highly interlinked in terms of mobility, but *even there* a subnational approach would have benefited the intervention strategy — which does create high expectations of similar approaches in other countries. Second, on a meta-level, the results yield an important message to policymaking to rethink how and at which level mandates are distributed across institutions in epidemic situations and to at least consider the potential benefits of providing regional institutions such as provinces or municipalities with the possibility to apply differentiated action. The results feed the more general discussion on the balance of societal impact of lockdowns and pressure on the health system. Third, in terms of methodology, the main merit of our approach lies in the fact that it captures the local context by coupling empirical data sources on demography, mobility, and spatial clustering of the population and link this to disease transmission. This makes the approach itself, rather than the specific numbers, exportable to other settings. Additionally, we show how to decouple individual interventions in *Figure 3*, which is made possible by capturing the local context: mobility reduction and behavioral changes cannot be separated if mobility and behavior are not explicitly modeled. (Even though we note that even at our high resolution level, there are limits to which such interventions can be fully distinguished, as mentioned above.) In Appendix 2.3, we have also added hypothetical scenarios on closing municipality borders, closing schools earlier and initializing the epidemic in Amsterdam — all of which can be studied using a framework such as that described here. Building on these points, we believe this paper adds to the discussion on intervention approaches in any future epidemic beyond the case study.

That said, whether subnational implementation of interventions is sufficient to successfully control an epidemic should be expected to vary across countries and cultures, as its effectiveness depends on its timeliness and feasibility, and it being sufficiently supported and/or enforceable. For example, subnational control measures worked (to some extent) in China due to high enforcement and feasibility (sheer man power), and in New Zealand and Australia due natural long distances between subpopulations and relative isolation from the rest of the globe (*Sachs et al., 2022*). In contrast, during the initial wave in Northern Italy which is a highly connected region, the regional lockdown was implemented too late to curb further geographical spread (*Waitzberg et al., 2022*).

## Materials and methods

This section is devoted to discuss a few of the core concepts of the methods. For a detailed step-by-step explanation, see Appendix 1.1-1.8.

### Agents and their mobility patterns

The basis for the mobility patterns is anonymized mobile phone signal data gathered by a commercial data provider, resulting in numbers of daily travels by people living in municipality $i$ to municipality $j$, split into frequent, regular and incidental movements. Additionally, the demographic data provided by Statistics Netherlands (CBS) allowed us to distinguish 170,721 agents (with roughly

17 million residents, this means that each agent represents about 100 of them) with demographic details (Appendix 1.1). For each agent, we determine movements by drawing from mobility distributions computed from the mobile phone signal data, in which we distinguish frequent from incidental and regular movements by making assumptions about the reasons of moving (work and school versus other activity). More specifically, the generated mobility distributions are Dirichlet distributions, using the (normalized) movements data as shape parameters. From these distributions, we independently draw fractions of the day spent in each municipality (i.e. resulting in 380 fractions for each of the 380 municipalities), that are subsequently converted into integer hours spent in municipalities. More detailed information on the computing of the agents' movements can be found in Appendix 1.2 and Appendix 1.3.

## Pathogen transmission

Transmission from susceptible ($S$) to exposed ($E$) in this stochastic SEIR-based model is based on a 'force of infection' $\lambda$, which is translated to an hourly infection probability. The idea behind $\lambda$ exerted on a susceptible agent is that each demographic category contributes to the chance of transmission of the pathogen to this agent, weighted by the expected mixing between the agent and this category, as well as on the fraction infectious in this category. The full equation for $\lambda$ for people from demographic group $g$ in municipality $m$ at time $t$, involving a summation over all demographic groups $g'$ adding to the force of infection, is as follows.

$$\lambda(g,m,t) = \underbrace{h(g)}_{\text{Susceptibility of } g} \cdot \underbrace{\beta_t \cdot \bar{s}(t)}_{\text{Phase \& daily cycle}} \cdot \underbrace{\sum_{\text{Group } g'} n_{g,g'} \cdot \frac{I(g',m,t)}{N(g',m,t)}}_{\text{Mixing with groups } g'} \cdot \tag{1}$$

The first part on the right hand side of the equation involves a parameter $h(g)$ that reflects the susceptibility of an agent belonging to demographic group $g$ to the disease (see Appendix 1.1). The second part ($\beta_t \cdot \bar{s}(t)$) contains the behavioral parameter $\beta_t$ (such as wearing face masks and maintaining social distance) depending on the phase of the wave (leading to $\beta_1$-$\beta_4$, see **Appendix 1—table 1**) and a daily cycle parameter $\bar{s}(t)$ (see Appendix 1.6); for example, ensuring that agents barely have any contacts in the middle of the night. The third part involves the mixing with the eleven different demographic groups: $n_{g,g'}$ is the expected number of contacts that group $g$ has with group $g'$, based on the mixing matrix that reflects the situation (i.e. 'home', 'school', 'work' or 'other'). The fraction $\frac{I(g',m,t)}{N(g',m,t)}$ is the fraction of the total number ($N$) of agents belonging to group $g'$ in municipality $m$ that are infectious ($I$).

The time scales of transitions from exposed ($E$) to infectious ($I$) and from infectious ($I$) to recovered ($R$) – expressed in an incubation and an infection time scale, respectively – differ per case and are drawn from Weibull distributions with mean time scales of 4.6 and 5 days, respectively (**de Vlas and Coffeng, 2021**; Appendix 1.5).

## National-level interventions

The first COVID-19 wave in the Netherlands lasted over the period February 27 (first reported case) to June 1, 2020. Based on the interventions that took place, we split this period into four phases, for

**Table 1.** Overview of how the four phases in the first wave of COVID-19 in the Netherlands are implemented in our analysis.

| Phase | Start | End | Travel | Mixing | Behavior | Schools |
|---|---|---|---|---|---|---|
| 1 | Feb 27 | Mar 11 | - | - | $\beta_1 = 0.135$ | Open |
| 2 | Mar 12 | Mar 22 | −31.7% | Reduced as per Apr 2020 | $\beta_2 = 0.11$ | Closed halfway* |
| 3 | Mar 23 | May 10 | −42.4% | Reduced as per Apr 2020 | $\beta_3 = 0.09$ | Closed* |
| 4 | May 11 | Jun 1 | −20.1% | Reduced as per Jun 2020 | $\beta_4 = 0.11$ | Open |

*Schools were closed in the period March 16 – May 10, which is also what we use in our analysis.

which we analyze the epidemiological impacts of changes in mobility, mixing, behavior and school closure. Details about these phases are shown in *Table 1*.

In our analysis, we capture these changes in the following manner. First, we reduce inter-municipality mobility as reported by *Google, 2021* in the four phases of the first wave in the Netherlands. The dominant contribution to this travel reduction, by far, was due to a working-from-home policy recommended by the Dutch government; we implement it in our analysis by placing the reported percentage of agents, randomly drawn from the working categories, at home. Secondly, we address changes in mixing patterns by determining percentage changes in the mixing among different age groups from Dutch survey data (*Backer et al., 2021*) in the months February, April, and June 2020, and applying these changes element-wise to the mixing matrices used in our analysis. Thirdly, we represent behavioral changes by variations in $\beta_t$ in *Equation 1* across the four phases of the first wave. Fourth and finally, we implement school closing by placing school-going agents (i.e. primary school children, secondary school children and students) as well as the parents of primary school children at home, both in terms of the home locations of the agents and in terms of its implications on mixing (see Appendix 1.5).

## Acknowledgements

The authors thank Mirjam Kretzschmar and Hans Heesterbeek for a careful reading of the manuscript.

Funding: The authors gratefully acknowledge financial support via ZonMw grant 10430022010001. In addition, LEC acknowledges funding from the Dutch Research Council (NWO, grant 016.Veni.178.023).

## Additional information

### Funding

| Funder | Grant reference number | Author |
| --- | --- | --- |
| ZonMw | 10430022010001 | Debabrata Panja |

The funders had no role in study design, data collection and interpretation, or the decision to submit the work for publication.

### Author contributions

Mark M Dekker, Conceptualization, Data curation, Software, Formal analysis, Validation, Investigation, Visualization, Methodology, Writing – original draft, Writing – review and editing; Luc E Coffeng, Frank P Pijpers, Conceptualization, Formal analysis, Supervision, Methodology, Writing – review and editing; Debabrata Panja, Conceptualization, Supervision, Funding acquisition, Writing – original draft, Project administration, Writing – review and editing; Sake J de Vlas, Conceptualization, Supervision, Writing – review and editing

### Author ORCIDs

Mark M Dekker http://orcid.org/0000-0002-3543-6889
Luc E Coffeng http://orcid.org/0000-0002-4425-2264
Frank P Pijpers http://orcid.org/0000-0001-7572-9435
Debabrata Panja http://orcid.org/0000-0003-2141-9735
Sake J de Vlas http://orcid.org/0000-0002-1830-5668

### Decision letter and Author response

Decision letter https://doi.org/10.7554/eLife.80819.sa1
Author response https://doi.org/10.7554/eLife.80819.sa2

## Additional files

### Supplementary files
• MDAR checklist

## Data availability

Data associated with mobility and mixing reductions (Google mobility and PIENTER) (*Google, 2021*; *Backer et al., 2021*), age-stratified mixing matrices used in the analysis (POLYMOD; *Prem et al., 2017*), and hospital admission data (NICE) publicly available as described in Appendix 1.5, have been made available at the Data Repository https://osf.io/muj4q/. All analysis codes have been made available at https://github.com/MarkMDekker/covid_intervention_evaluation (copy archived at *Dekker, 2022*). Our analysis also uses mobility information as input. This dataset is owned by a commercial party (Mezuro) and can therefore not be made public. For the purpose of enabling readers to run our codes and obtaining comparable results, we have made synthetic mobility data available, also at the Data Repository https://osf.io/muj4q/. This synthetic data has been generated using a gravity model. For frequent travels, this is entirely standard, for infrequent visits square root of the distance is used in the numerator. The prefactor in the standard gravity model is chosen as 0.5 to account for the double counting due to return journeys. For infrequent visits, mostly weekend trips, we have used G=1/7. Request for the actual mobility data can be sent to info@mezuro.com as a proposal. Access to the data may require payment, and will certainly be subject to vetting related to privacy issues by GDPR (General Data Protection Regulation).

The following dataset was generated:

| Author(s) | Year | Dataset title | Dataset URL | Database and Identifier |
| --- | --- | --- | --- | --- |
| Dekker M | 2022 | Reducing societal impacts of SARS-CoV-2 interventions through subnational implementation | https://osf.io/muj4q/ | Open Science Framework, muj4q |

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

# Appendix 1

## 1. Analysis framework components

This Appendix is devoted to explaining the eight steps in *Figure 1* in greater detail.

We start with 170,721 agents divided into 11 demographic groups. Following the demographic population distribution data obtained from the Central Bureau of Statistics (CBS) of the Netherlands, we assign them proportionately to (i.e. as living in) one of the 380 municipalities in the country. This defines the 'home municipality' and the demographic group for each agent. Based upon this, and using mobility data accumulated by a commercial data provider from mobile phone signals in 2018, we stochastically infer every agent's movements throughout the country (steps 1–4 in *Figure 1*). Once we have determined the locations of every agent on an hourly basis in this manner, we simulate an individual-(agent-)based SEIR model (steps 5–8 in *Figure 1*).

### 1.1 Agents and demographic groups [Step 1]

Step 1 concerns the definition of the agents and their attributes. By choosing to represent the Dutch population by 170,721 agents, we essentially adopt 1:100 population scale, meaning that each agent in the model represents 100 people in the population. According to CBS, the Dutch population in 2019 was 17,256,870. While this means that we should ideally have a total of 172,569 agents in our analysis, we end up with 170,721 agents because of rounding, since we need an integer number of agents living in each municipality.

The characteristics of agents are set using real demographic data, but not directly one-to-one at an individual level ('micro-level'), because this would not conform to the standards for protection of sensitive data imposed by the legal requirements around the use of CBS data. First, micro-level data are aggregated to a municipal level through automated procedures internal to CBS and not accessible to outside researchers. For research purposes instead a distribution function is then provided for any given characteristic or attribute. Subsequently, using a Monte Carlo process, as many samples can be drawn from this distribution as necessary to provide the required synthetic population of agents for that municipality.

In the Netherlands, municipalities are merged or land areas are reassigned between them on a fairly regular basis, with the aim of making local governance more effective. Most years, the exact subdivision of the country into municipalities therefore changes a little bit. Year-to-year changes of municipality divisions are minor and only affect small municipalities. This does mean that the mobility data and the demographic data, which are collected in different years, need to be transformed slightly, so that they can be made to correspond to identical geographical divisions. Because the mobility data (see Appendix 1.2) was using the municipality division of 2018, we projected the demographic data from 2019 onto the municipalities as they were in 2018. (Note that in turn, purely for plotting purposes (e.g. for the maps in *Figure 1*), the municipality borders and shape files of 2020 are taken, requiring an additional projection when visualizing the results.)

For every municipality, we split the agents living therein into eleven demographic categories, keeping track of, e.g., whether they go to school or not, study, or their employment status. Data on whether agents go to school or their employment status etc. are obtained from tax records and education institutional registers, also provided by CBS. Distinguishing criteria other than age provides additional information on how the agents move across municipalities (frequent movements or otherwise) and mix (*Mossong et al., 2008*). For our analysis, this information allows us to more explicitly target the right agents for implementing intervention measures (see Appendix 1.7-1.8).

Relevant information on each demographic group is displayed in *Appendix 1—table 1*. The Middle-age working category is by far the largest. Note that these fractions are not constant across municipalities: some Dutch municipalities, especially those in the north and east of the country, have higher-than-average amounts of elderly and fewer children, and the opposite holds for larger cities in the west.

**Appendix 1—table 1.** The eleven demographic groups and attributes relevant to our analysis.
The mixing situations in the last two columns point towards the matrices chosen to simulate the mixing (see Appendix A.4): without brackets if the agent is in the home municipality, in brackets if the agent is in a different municipality.

| Group | Criteria | | | | Attributes | | |
| --- | --- | --- | --- | --- | --- | --- | --- |
| | Age (y) | Work | School | National total (fraction) | Av. time not in home municipality | Daytime mixing | Nighttime mixing |
| Pre-school children | 0–4 | - | - | 851880 (4.9%) | 6% | Home (Other) | Home (Other) |
| Primary school children | 5–11 | - | Yes | 1295380 (7.5%) | 6% | School | Home (Other) |
| Secondary school children | 12–16 | - | Yes | 991290 (5.7%) | 6% | School | Home (Other) |
| Students | 17–24 | - | Yes | 1,086,240 (6.2%) | 26% | School +Work | Home (Other) |
| Non-studying adolescents | 17–24 | - | - | 632,530 (3.6%) | 26% | Work | Home (Other) |
| Middle-age working | 25–54 | Yes | - | 5,530,360 (31.8%) | 26% | Work | Home (Other) |
| Middle-age unemployed | 25–54 | - | - | 1,231,780 (7.1%) | 6% | Home (Other) | Home (Other) |
| Higher-age working | 55–67 | Yes | - | 1,623,040 (9.3%) | 26% | Work | Home (Other) |
| Higher-age unemployed | 55–67 | - | - | 1,109,170 (6.4%) | 6% | Home (Other) | Home (Other) |
| Elderly | 68–80 | - | - | 2,102,530 (12.2%) | 6% | Home (Other) | Home (Other) |
| Eldest | 80+ | - | - | 802,670 (4.6%) | 6% | Home (Other) | Home (Other) |

## 1.2 Mobility across municipality borders [Step 2]

Given the distribution of the agents and the demographic categories across the Netherlands, we need an empirical basis for normal (i.e. pre-pandemic) inter-municipality mobility patterns of the agents. The data is provided by a commercial data provider 'Mezuro' (henceforth called 'Mezuro data' and is further elaborated on in Appendix 3.2) over the period of March 1, 2019 up to and including March 14, 2019. Specifically, the data comprises of a matrix $M_{ij}$ showing how many (daily average) visits there were from people living in municipality $i$ to municipality $j$. The data does not contain information on movements within a given municipality. Neither does it provide information on sequence of movements — for example when a person living in Amsterdam travels to Rotterdam and afterwards travels to Utrecht, Mezuro data registers it as if there is a movement of an Amsterdam inhabitant to Rotterdam, and a separate movement of an Amsterdam inhabitant to Utrecht (i.e. the in-between stop of this person at Rotterdam is not registered). In the Mezuro data, there is no information on the duration of anyone's stay in any given municipality, nor is there any information about demographic aspects of people's mobility.

The daily average visits in $M_{ij}$ are however split into three categories: 'frequent', 'regular', and 'incidental' (for more information, see Appendix 3.2). We use this information to crudely infer which mobility data to use for which demographic category: We apply frequent and regular movements to working and school-going agents, while we apply incidental movements to all other agents. In other words, we use two mobility matrices: ($M_{freq}$) describing regular and frequent movements, and ($M_{inc}$) describing incidental movements, both averaged over the 14 days in the period March 1 – March 14, 2019.

## 1.3 Agent movements [Step 3]

The third step of the model concerns linking the mobility data to the movements of individual agents in the model. The movements are determined per agent. Each agent belongs to a particular demographic group $g$ and lives in a municipality $m$. Given demographic group $g$, we use either $M_{freq}$ or $M_{inc}$, as mentioned above. In the following, we use $M$ to denote the choice of one of these matrices. The municipality $m$ points us to the particular row $M_m$ to use from the mobility matrix, containing the number of movements from $m$ to other municipalities. We use this information to create the scale parameters for the Dirichlet distribution, from which fractions of the day are drawn that the agent (living in $m$, belonging to $g$) spent in each municipality. We do this as follows.

First, we have to normalize these movements in a proper manner. To account for the fact that people living in some municipalities have above-average amount of movements, we do not normalize the movements $M_m$ by the total (i.e. resulting in a row-sum of 1), but by the amount of people living in $m$, effectively obtaining the amount of movements to each other municipality *per inhabitant of m*. However, because we aim to use these elements to eventually draw fractions of the day spent in each municipality, we also need to set how much time is spent in the home municipality itself, as the data only contains movements between municipalities. We assume that working people and

students spend approximately 25% of their time, and not-working people spend approximately 5% of their time in other municipalities. Using the average row sums divided by the populations as mentioned above, this results in values of 1 and 1.5, respectively, for weighting the time spent in the home municipality. Summarized, this boils down to the following expression of the $m'$th scale parameters $\delta$ of the Dirichlet distribution, for people living in municipality $m$, belonging to working people and students:

$$\delta(m, m') = \begin{cases} \frac{M_{\text{freq},mm'}}{P(m)} \cdot \frac{2.5}{\sum_i \delta(m,i)} & \text{if } m \neq m' \\ 1 \cdot \frac{2.5}{\sum_i \delta(m,i)} & \text{if } m = m' \end{cases} \quad (2)$$

where $P(m)$ is the population of municipality $m$. And analogously for other demographic groups:

$$\delta(m, m') = \begin{cases} \frac{M_{\text{inc},mm'}}{P(m)} \cdot \frac{2.5}{\sum_i \delta(m,i)} & \text{if } m \neq m' \\ 1.5 \cdot \frac{2.5}{\sum_i \delta(m,i)} & \text{if } m = m' \end{cases} \quad (3)$$

The factor $\frac{2.5}{\sum_i \delta(m,i)}$ sets the total sum of the Dirichlet parameters to 2.5, to set the variability between draws from the resulting distribution to be constant. Histograms of the row sums of the mobility matrices, divided by the municipality population sizes are shown in *Appendix 1—figure 1*. The resulting Dirichlet distribution is used to draw fractions of the day that a person belonging to the respective demographic group and living in the respective municipality spends in each municipality — meaning that, per person, 380 fractions are drawn, one for each municipality. Because of the definition of the shape parameters, the largest fraction is usually his/her home municipality. These fractions are converted to integer-hours by multiplying with 24 and rounding down (leftover hours are spent in home municipalities).

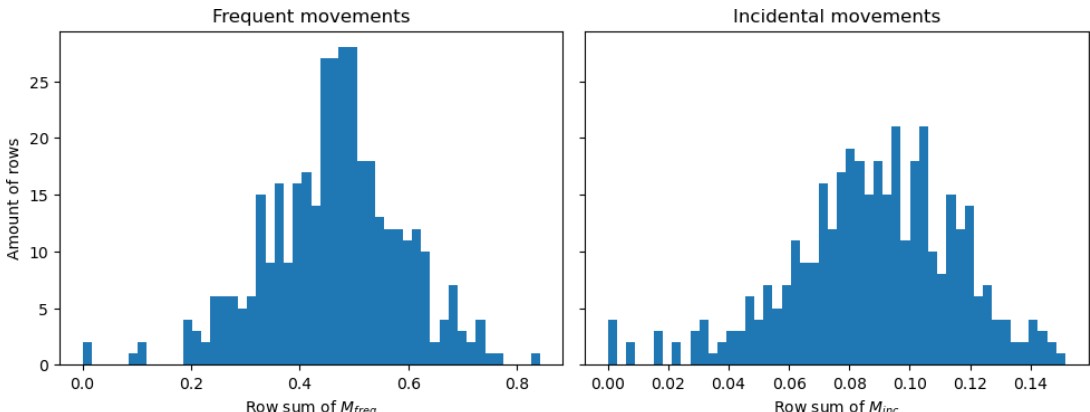

**Appendix 1—figure 1.** Histograms showing the row sums of (**a**) matrix $M_{\text{freq}}$ and (**b**) matrix $M_{\text{inc}}$ divided by the population sizes belonging to those rows.

To convert this list into an actual schedule of this person on this particular day, the order of these visits should be decided. The time spent in the home municipality is cut in two and the halves are placed at the beginning and ending of the day, marking staying at home overnight. For example, given that $p$ spends 14 hr of the day in their home municipality, then $p$ is assumed to spend the hours 00:00-07:00 and 17:00-24:00 in this municipality. Duplicate hours spent in other municipalities are concatenated and these concatenated periods are place in the leftover hours in a random order. This leads to daily 'schedules' such as visualized in the bottom panel of Step 3 in *Figure 1*. The motivation for working on an hourly resolution stems from the fact that sequence of movements are highly important for epidemiological spreading: the fact that people meet during the day in municipality $A$ for a short while can make a large difference already.

We repeat this procedure seven times to end up with a weekly schedule. This ultimately results in a movement pattern that varies from day to day, but is the same for each day of the week (e.g. Monday in week 1 are equal to Mondays in other weeks). The resulting fraction of people in other

(not-home) municipalities is shown in *Appendix 1—figure 2* for different moments of the day, having an average of 16.6% of the time spent in other municipalities.

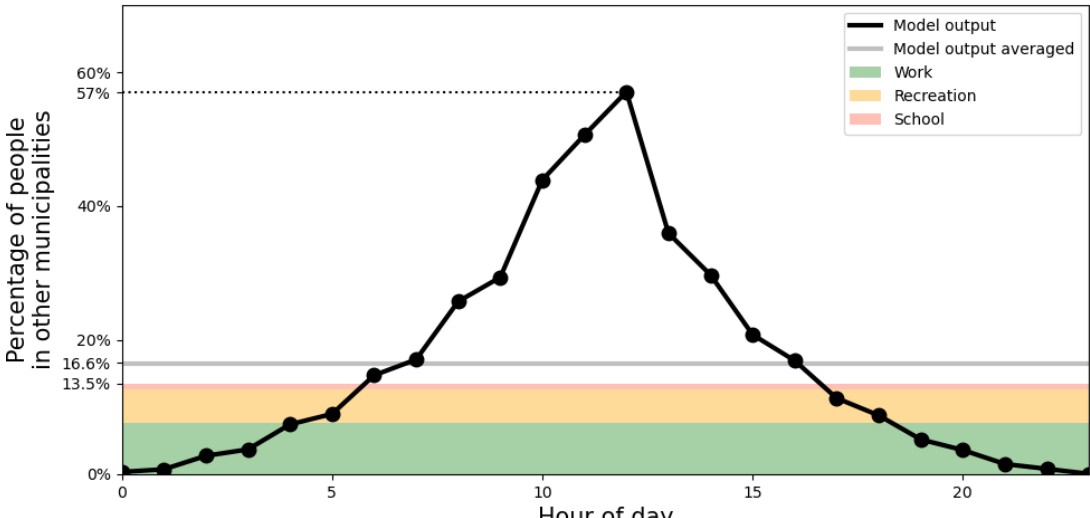

**Appendix 1—figure 2.** Percentage of people outside of their home municipality (black). At the bottom, in colors, daily-averaged estimates of time spent outside the home municipality are displayed, due to the activities of work (green), recreation (yellow), and school (red).

We check the approximate validity of this pattern by comparing these results to survey data from Dutch governmental research agencies. In particular, from a regular survey done by the Sociaal Cultureel Planburea (SCP), we know that people (12 years old and older) in 2016 spent on average 20.5 hr on paid work, 3.3 on schooling and 42.1 on recreation (*Social Cultureel Planbureau, 2016*) per week. Also, 38% of the people lives in the municipality that they work in *Centraal Bureau voor de Statistiek, 2021*, that is, 62% has to travel to another municipality for work. This means that $\frac{20.5}{7\cdot24} \cdot 62\% = 7.6\%$ of the total time of a day is on average spent in other municipalities because of work — the division by 7.24 is to convert from week to daily numbers. Furthermore, we know that 48% of students do not live in dorms (*Kenniscentrum Studentenhuisvesting, 2020*), which we can use to approximate how many students have to travel between municipalities for their schooling. Analogous to working time, we reason that an additional fraction of time spent outside of the home municipality due to schooling is $\frac{3.3}{7\cdot24} \cdot 48\% = 0.94\%$, assuming that students living in dorms outside of their school municipality and the fact that youth between 12 and 18 do not live in dorms partially counterbalance. Concerning recreation, we assume 20% of recreation being outside of the home municipality, which results in yet another additional fraction of $\frac{42.1}{7\cdot24} \cdot 20\% = 5.0\%$. Summing them results in 13.5%. This is less than the observed 16.6% (gray line in *Appendix 1—figure 2*), but there are many large uncertainties in these calculations (e.g. the time spending survey (*Social Cultureel Planbureau, 2016*) is only based on people of 12 years and older), but we use them to have an approximate validation.

Another validation to be made concerns the factor 2.5 in *Equations 2; 3*, which is the total sum of the parameters of the Dirichlet distributions. Statistically, the total sum of the scale parameters in a Dirichlet distribution indicates the variability across draws. Therefore, we need to make sure that the variability we set here makes sense. We do this by calculating the fraction of the day spent in the home municipality of people, and visualize the between-people variability in this metric. The results are shown in *Appendix 1—figure 3*. In the left panel, the near-separation is noticeable between groups that had increased cross-municipality movement (i.e., home-scale parameter of 1 in the Dirichlet distribution; students and working people) and those that did not (i.e. parameter of 1.5; other groups). Also, we see that in many groups (also in the 'All' category), the distributions of how long people are in their own municipality varies, and the tails overlap. The differences across the municipalities in the right column are explained by differences in their mobility-to-population ratio (illustrated in the row sums in *Appendix 1—figure 1*) and demographic differences. Even though we do not have observed data to compare these numbers, they do not seem to be unrealistic: large

cities such as Utrecht, Amsterdam, and Rotterdam involve people that are probably working there, and may indeed therefore have a higher fraction of time spent in their home municipality.

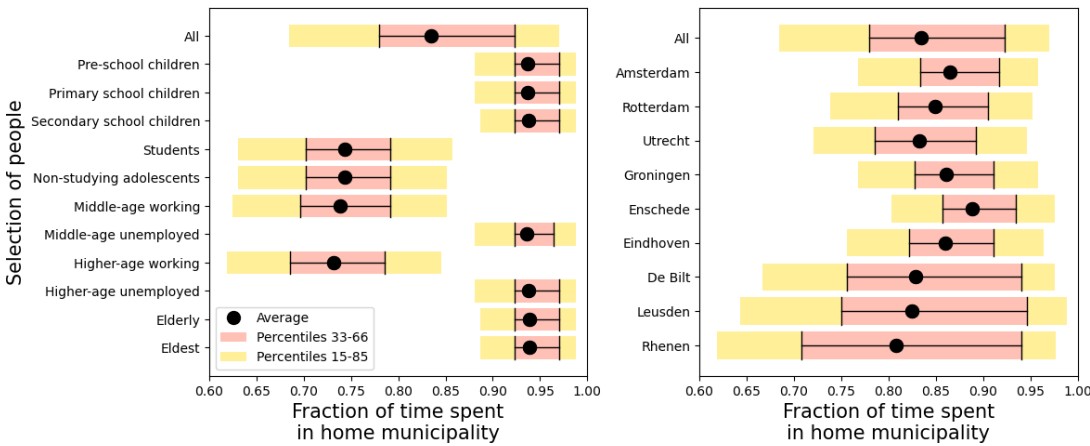

**Appendix 1—figure 3.** Variability in the fraction of time spent in the home municipality. Panel (**a**): across different demographic groups. Panel (**b**): across a selection of municipalities.

## 1.4 Mixing [Step 4]

From steps 1–3, we know which people are in the same municipality at which moment of time. In step 4, we determine the mixing of different demographic groups *within* a municipality, assuming proportionate mixing. We base this on two factors: a demographic stratification in the mixing, and a distinction of four different mixing situations. We follow the POLYMOD study (*Mossong et al., 2008*; *Prem et al., 2017*) by distinguishing four unique situations: 'home', 'school', 'work', and 'other'. From *Prem et al., 2017*, we downloaded the average contact rate matrices, and converted them from their 16 demographic groups to our 11 demographic groups.

Which of the four situations applies to a person, is determined by time of day, whether the person is in his/her home municipality, and the demographic category the person belongs to. The exact mixing matrices used are shown in *Appendix 1—table 1*, under 'Daytime mixing' (corresponding to time between 8 and 18) and 'Nighttime mixing' (other times of day). For the Students group, we make an exception by not taking one specific mixing matrix, but averaging the 'work' and 'school' mixing matrices during daytime.

## 1.5 Hospital admissions and model initialization [Step 5]

### Hospital admissions

The hospital admission data is obtained from the Nationale Intensive Care Evaluatie (NICE) registration, which is the official institution for hospital reports. In particular, the data can be found under https://data.rivm.nl/meta/srv/dut/catalog.search, which are daily numbers per municipality. In particular, this means that all model results had to be translated into daily numbers (summing over a moving window of 24 time steps).

We proceed by discussing the translation between infection cases and hospital admissions, which is required for both the initialization and calibration. Following *de Vlas and Coffeng, 2021* we use a time lag between becoming symptomatic (in model terms: infectious $I$) and a potential hospital admission of being Weibull-distributed with mean 14 and scale parameter 10 (*de Vlas and Coffeng, 2021*). The probability of hospital admission $p_{hos}$, given that a person becomes infectious is not equal for all people: elderly are more susceptible to being hospitalized than young children, for example. In *Appendix 1—table 2*, we show $p_{hos}$ across the various demographic groups. These probabilities are determined as follows. Using the seropositivity data during the Dutch first wave (*Vos et al., 2020*) and knowing how many people there are in each age group (see Appendix 1.1), we can estimate the cumulative amount of infection cases per group. Combining this with cumulative hospital admission data per age group, we can divide the two to get the hospitalization probability per age group, which can be translated to the 11 demographic categories we use. These probabilities $p_{hos}$ per demographic category are shown in *Appendix 1—table 2*. The Weibull-distributed temporal

translation and demography-stratified probability of hospitalization are done when translating the model results into hospital admissions, for example in the red curve in *Figure 2(a)*.

**Appendix 1—table 2.** Probability of hospital admission $p_{hos}$ and susceptibility parameter $h(g)$.

| Demographic group | $p_{hos}$ | $h(g)$ |
|---|---|---|
| Pre-school children | 0 | 1.0 |
| Primary school children | 0 | 2.0 |
| Secondary school children | 0.0018 | 3.051 |
| Students | 0.0006 | 5.751 |
| Non-studying adolescents | 0.0006 | 5.751 |
| Middle-age working | 0.0081 | 3.6 |
| Middle-age unemployed | 0.0081 | 3.6 |
| Higher-age working | 0.0276 | 5.0 |
| Higher-age unemployed | 0.0276 | 5.0 |
| Elderly | 0.0494 | 5.3 |
| Eldest | 0.0641 | 7.2 |

## Model initialization

The model is initialized with an estimated amount of infectious cases in the period up to March 1, 2020. Initializing with fewer cases (i.e., up to before March 1) would increase the sparsity of initial cases, which especially when working on a 1:100 resolution may result in high stochasticity (e.g. when in some crucial municipalities the disease suddenly dies out). Taking a longer initialization input (i.e. up to later than March 1) would limit our ability to test intervention measures in Phase 1. This had to be derived from hospital admissions, because the testing capacity was so low in this period that the tested infection cases cannot be used to estimate the real number of infections.

While the conversion of model output to hospital admissions is done using a Weibull distributed time lag, we do the initialization simpler. We start with hospital admission data in the observed data (i.e. it is the other way around), which are specified per municipality. We use a flat percentage $p_{hos,0}$, which is the weighted average of all hospital admission probabilities in *Appendix 1—table 2*, weighted by the respective category size, and we translate these numbers back 14 days into the past (instead of fully Weibull-distributed). The respective affected agents are randomly drawn (within each specified municipality) based on the resulting numbers. These were assigned the disease stage 'infectious' $I$, while the rest of the population was assigned 'susceptible' $S$. We aim to approximate in the initialization the amount of infection cases up to March 1, which means we have to take hospital admission data up to March 15.

## 1.6 Disease transmission and force of infection [Step 6]

The transmission dynamics and force of infection $\lambda$ are already discussed in the Methods section. This section focuses on the susceptibility parameter and the daily cycle, both part of the equation for $\lambda$. The first part of the equation involves a parameter $h(g)$ that reflects the susceptibility to the disease, based on the demographic group $g$. The age-specific susceptibility parameter shown in *Appendix 1—table 2* is based on estimations of previous work (*RIVM, 2021b*), table A3. The second part $[\beta_t \cdot \bar{s}(t)]$ contains a parameter $\beta_t$ that involves behavioral aspects like wearing face masks and keeping social distance — this parameter is used for distinguishing four phases in the epidemic as described later. The parameter $\bar{s}(t) = \frac{s(t)}{\sum_t s(t)}$ (where $t \in [0, 23]$ in hours) is what we refer to as the 'daily cycle parameter', reflecting the fact that people hardly mix during the night, and more throughout the day (see for values of $s(t)$ per hour in *Appendix 1—table 3*).

**Appendix 1—table 3.** Daily cycle parameter $s(t)$, from which we obtain $\bar{s}(t)$.

| Hour of day ($t$) | 0 | 1 | 2 | 3 | 4 | 5 | 6 | 7 | 8 | 9 | 10 | 11 | 12 | 13 | 14 | 15 | 16 | 17 | 18 | 19 | 20 | 21 | 22 | 23 |
|---|---|---|---|---|---|---|---|---|---|---|---|---|---|---|---|---|---|---|---|---|---|---|---|---|
| $s(t)$ | 1 | 1 | 1 | 1 | 1 | 1 | 0.75 | 0.5 | 0.25 | 0 | 0 | 0 | 0 | 0 | 0 | 0 | 0 | 0 | 0 | 0.2 | 0.4 | 0.6 | 0.8 | 1 |

## 1.7 Intervention data sources [Step 7]

There are four factors that we apply to mimic intervention measures, spread across four phases, summarized in *Table 1* in the main text. The first are behavioral changes such as wearing face masks and social distancing, represented by the values of $\beta_t$, varying across the four phases, yielding $\beta_1$-$\beta_4$. We use the $\beta_t$'s as calibration parameters, see Appendix 1.8.

Other simulated intervention measures were related to mobility, for example restricting various events and applying a working-from-home policy, which were all informed by data. The reduction in inter-municipality travel was quantified using Google Mobility data, which describe how mobility changed across this period across six categories (shown in *Appendix 1—figure 4*). Using the average of three of these categories — transit stations, workplaces and retail & recreation, chosen because these reflect inter-municipality travel best — and averaging the mobility changes within each phase, we end up with three scalar percentages for phases 2, 3, and 4, representing a mobility reduction. These percentages are implemented by randomly selecting the respective percentage of the population among the employed demographic categories, and placing them at home.

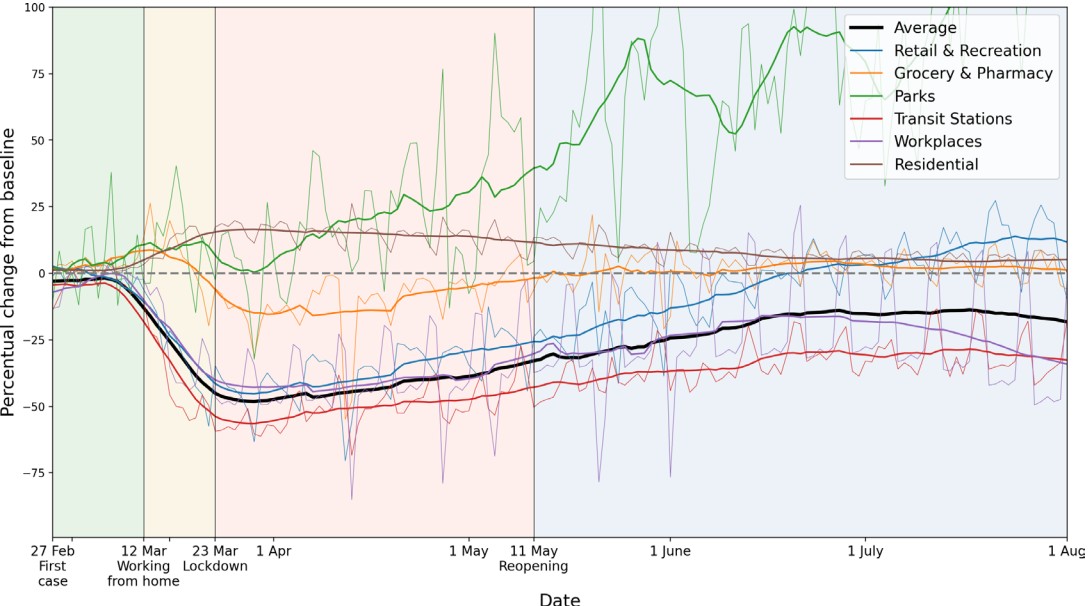

**Appendix 1—figure 4.** Percentage mobility changes with respect to the baseline, provided by Google Mobility data. Six categories are distinguished, in different colors. For the purpose of measuring how inter-municipality travel reduced, we use the average of the categories 'Transit Stations', 'Workplaces' and 'Retail and Recreation'. For each color, the raw data (thin lines) and 14-day running averaged data (thicker lines) are shown.

Across the first wave of COVID-19, people also started mixing differently. This is explicitly measured in the Netherlands using surveys in the PIENTER study (*Vos et al., 2020*). There, three survey studies are done: one in February 2020, one in April 2020 and one in June 2020. We only use this data in terms of their percentage changes: that of April with respect to February, and that of June with respect to February. The authors did not distinguish the same four unique mixing situations (home, work, school and other) as we use here, so we translate their (age-stratified) mixing changes into a single 11-by-11 matrix representing percentage changes in the contact rates between the demographic categories, and apply these percentage changes to all four mixing matrices in the same manner. In particular, the mixing changes in phase 2 and 3 are determined by the percentage mixing changes from the April surveys with respect to February surveys, and analogously, the changes in phase 4 are determined from the June surveys. See *Appendix 1—figure 5*.

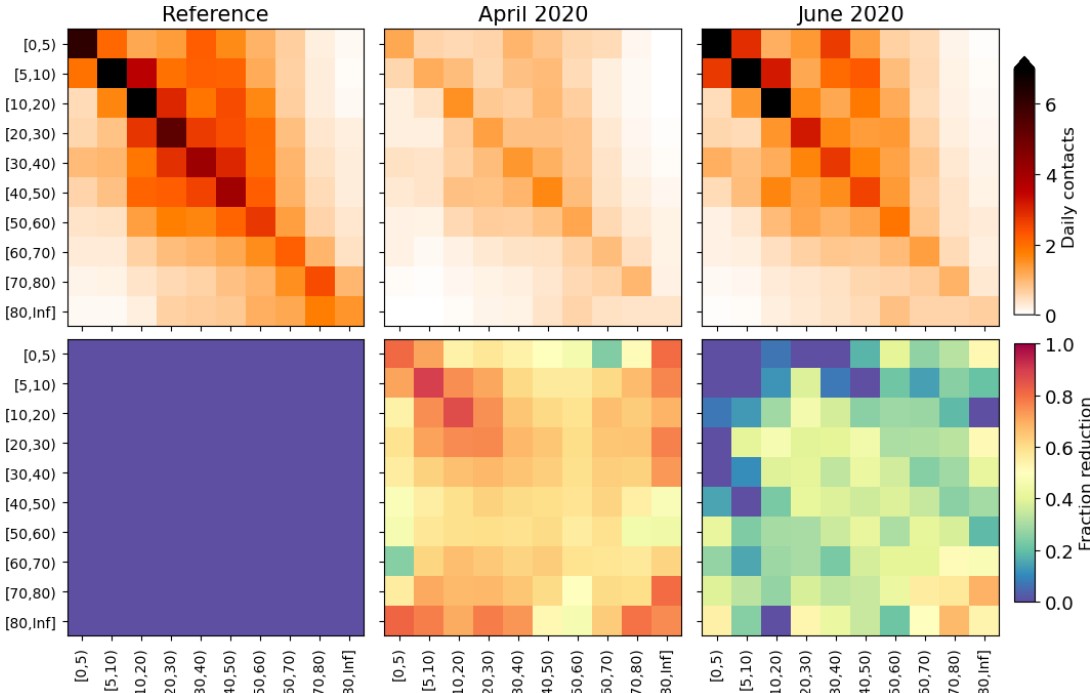

**Appendix 1—figure 5.** Changes applied to the model mixing matrices, calculated from survey mixing matrices in the PIENTER study (*Vos et al., 2020*). Top panels: daily contacts in absolute numbers, stratified by age as depicted in the reference study (left), April 2020 surveys (middle) and June 2020 surveys (right). Bottom panels: element-wise reductions in contact numbers relative to the reference. These reduction percentages are converted to the demographic groups we use in this study and applied throughout the phases.

A final important relevant factor to address among the changes and interventions during the first wave, was the closing of schools. The specific school closure dates were March 16, 2020 - May 10, 2020, which are directly used in the model, which in particular means that the schools get closed halfway through phase 2. School closure is implemented by two aspects. First, during daytime, all agents belonging to the demographic categories *Primary school children*, *Secondary school children* and *Students* are placed at their home municipality, and we now utilize the 'home' mixing matrix instead of the 'school' mixing matrix to determine their mixing. Second we incorporate the effect of parents of primary school children being forced to stay at home because their children are not going to school anymore. This means that, also during daytime, we place 12% of the *Middle-age working* people in their home municipality and set their mixing to 'home'. These are chosen separate from people working at home due to the mobility changes, to prevent double-counting.

The 12% is calculated as follows. In the Netherlands, 84% of people around 45 have children (*CBS, 2004*). Applying the assumption that agents in the Middle-age working group (25–54 years) have equal amount of children, we deduce that the ages of those children are uniformly distributed between −6.6 and 22.4 years old (from which you clearly see why we only use the Middle-age working category), using the average age of new mothers, which is 31.6 (*CBS, 2021*). Using the Dutch primary-school ages of 4–12 years, this means that $8/(22.4 + 6.6) = 28\%$ of these children are at primary schools. Assuming a rough estimate of 50% of those parents actually staying home (the rest having babysitters, family members or other means of taking care of their children), we end up with $0.84 \cdot 0.28 \cdot 0.50 = 0.12$, which is 12%.

## 1.8 Connecting model runs to real dates and calibration [Step 8]

Even though the model is initialized with an approximation of the infection cases up to March 1, each simulation requires a spinup to start mimicking the observed data well. This initial evolution is different for each simulation, and therefore we have to link each simulation separately to real dates. In other words, the model output has to be calibrated to actual dates. For this, we use the onset of phase 2 as a reference point. In particular, in each time step of the model simulation, we calculate the total amount of $I$ and $R$ agents. If this crosses the threshold of 1.8% of the population, phase 2 starts,

meaning that this is March 12. Once this is done, the calibration using the $\beta_t$ values could be done. The primary calibration is performed with $\beta_1$, to follow the initial increase in hospital admissions in *Figure 2*, as well as the observed doubling time. For $\beta_2$-$\beta_4$, we mainly focus on following to the observed hospital admission levels, as well as the qualitative fact that governmental policy on this was issued from phase 2 and on, being more strictly adhered to in phase 3, and loosened in phase 4. In other words: $\beta_t$ should decrease a bit in phase 2, even further in phase 3, and increase again in phase 4.

## 1.9 Subnational interventions

The subnational interventions tested in this study are analogous to national interventions in terms of timing and content, but whether they are in place is decided based on the local prevalence of infection: if the prevalence at a time $t$ has exceeded a threshold fraction of the municipality's population (i.e., 3%, 1%, 0.33%, or 0.1% as used in *Figure 4*), then at time $t$ exactly those interventions that were applied nationally are applied to the municipality. This leads to a proper comparison between the subnational and the national approaches.

Implementation-wise, the above means that changes to mixing and behavior (in the form of $\beta_t$ variations) are applied to people *present* (at any time) in the particular municipality, while the closure of schools and mobility reductions are applied to *inhabitants* of the municipality.

## Appendix 2

## 2. Additional model results

### 2.1 Explanation colors in *Figure 1*

In *Figure 1*, step 2, the green colors in the map and widths of the lines are both showing the same data, i.e., the total amount of visitors from Amsterdam to other municipalities on March 1, 2019, as marked by the mobility data (see Appendix 1.2). Several high-visited municipalities are highlighted by using black contours. In *Figure 1*, step 5, the total number of hospital admissions are shown between February 27 and March 15 in blue shades. Several municipalities with high hospital admission counts are highlighted using black contours. This information is used for the initialization, as mentioned in Appendix 1.5.

### 2.2 Seroposivity across demographic groups

*Appendix 2—figure 1* shows national seropositivity levels across the 11 demographic categories in different sets of national interventions. Although we add observed seropositivity (in black) to this panel, it is important to note that these values are highly uncertain because of a variety of biases involved in the data collection [**Vos et al., 2020**]. We add them to do a comparison of the general tendency across the demographic groups, which is higher seropositivity for adolescents, lower for older people and very low for the youngest — a tendency also found in the model output of the reference (green). Comparing the four scenarios reveals the same hierarchy as in panel (**a**), with a disproportionately high level of seropositivity for non-studying adolescents and middle-age working agents when travel reductions are omitted (red).

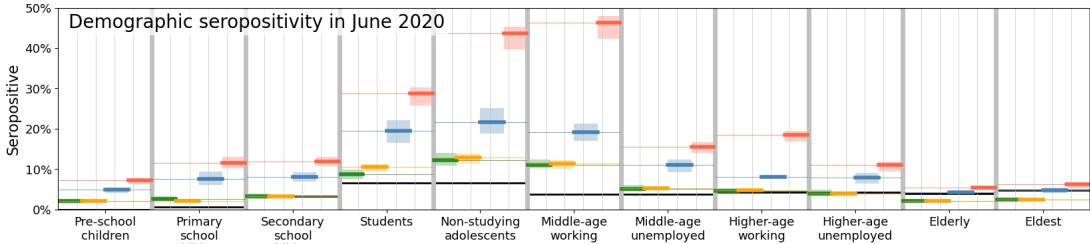

**Appendix 2—figure 1.** Comparison of disease prevalence across the eleven demographic groups (horizontal), expressed in % seropositivity (vertical) in different national administered intervention measures. Observed data are shown in black, the reference in green, and the impacts of (i) no behavioral changes like wearing masks, enhanced hygiene and social distancing in blue, (ii) no mobility reduction in red and (iii) no closing of schools in yellow. Bandwidths indicate the minima and maxima around the mean of a simulation ensemble of 40 realizations.

### 2.3 Other scenarios

The analysis framework can be used not only to evaluate subnational approaches when keeping the type of interventions synchronous to the (real) national interventions. In *Appendix 2—figure 2*, we show a few potential other scenarios (in gray and purple). In grey, we show what happens if municipality borders are closed upon initializing lockdown (i), and in purple, we show we show what happens if school are closed from the start of the simulation (ii). In scenario (i) (grey), when a threshold of local disease prevalence of 1.8% is reached, no agents can move in or out of the municipality anymore. Specifically, this means that agents *living* in the municipality will not move to other municipalities anymore, and agents that would normally move towards the specific municipality (e.g. to work), would stay in their home municipality, instead. Mixing, behavioral and school changes are similar to the Reference scenario (i.e. issued on a national level). Note that the threshold of 1.8% can be varied, analogous to the 3%, 1%, 0.33%, and 0.1% levels of subnational interventions in *Figure 4* in the main text. In scenario (ii) (purple), the simulations indicate that early school closure indeed reduces the amount of cumulative hospital admissions: on average by 13%.

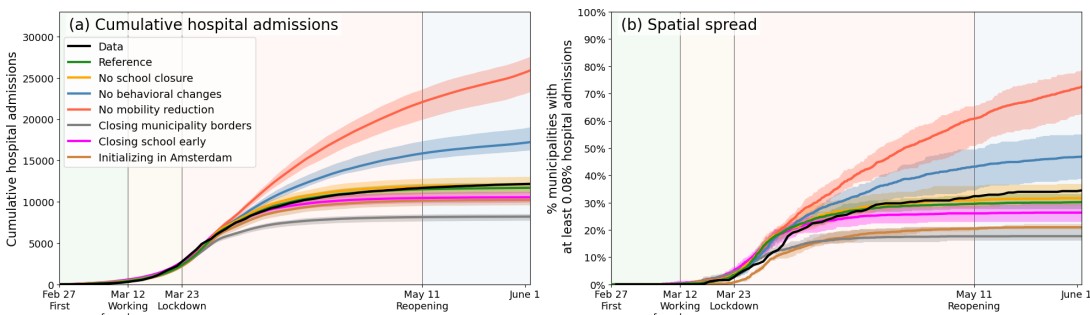

**Appendix 2—figure 2.** Comparing the impacts of nationally administered intervention measures. Same as *Figure 3* of the main text, but including three additional scenarios, in which municipality borders are fully closed upon initializing lockdown (gray), when schools are closed at the start of the simulation (purple), and when the outbreak starts in Amsterdam rather than as it was in reality (brown), respectively.

Finally, we add a third potential other scenario, in brown, in which all initial cases (normally spread in the south of the country) are found in the municipality of Amsterdam. Being the capital, this is a highly populous and interconnected area and the resulting disease spread is faster. However, in many ways, the outcome of this scenario is related to those of the scenario of closing schools early (purple), both in total hospital admissions (left) as well as in spatial spread (right).

## 2.4 Variability in geographic evolution within the ensembles

In *Appendix 2—figure 3*, we show the geographical distribution of the percentage of infectious and recovered on May 25 for 20 individual runs of the Reference scenario (i.e. the green lines in *Figures 3 and 4* of the main text).

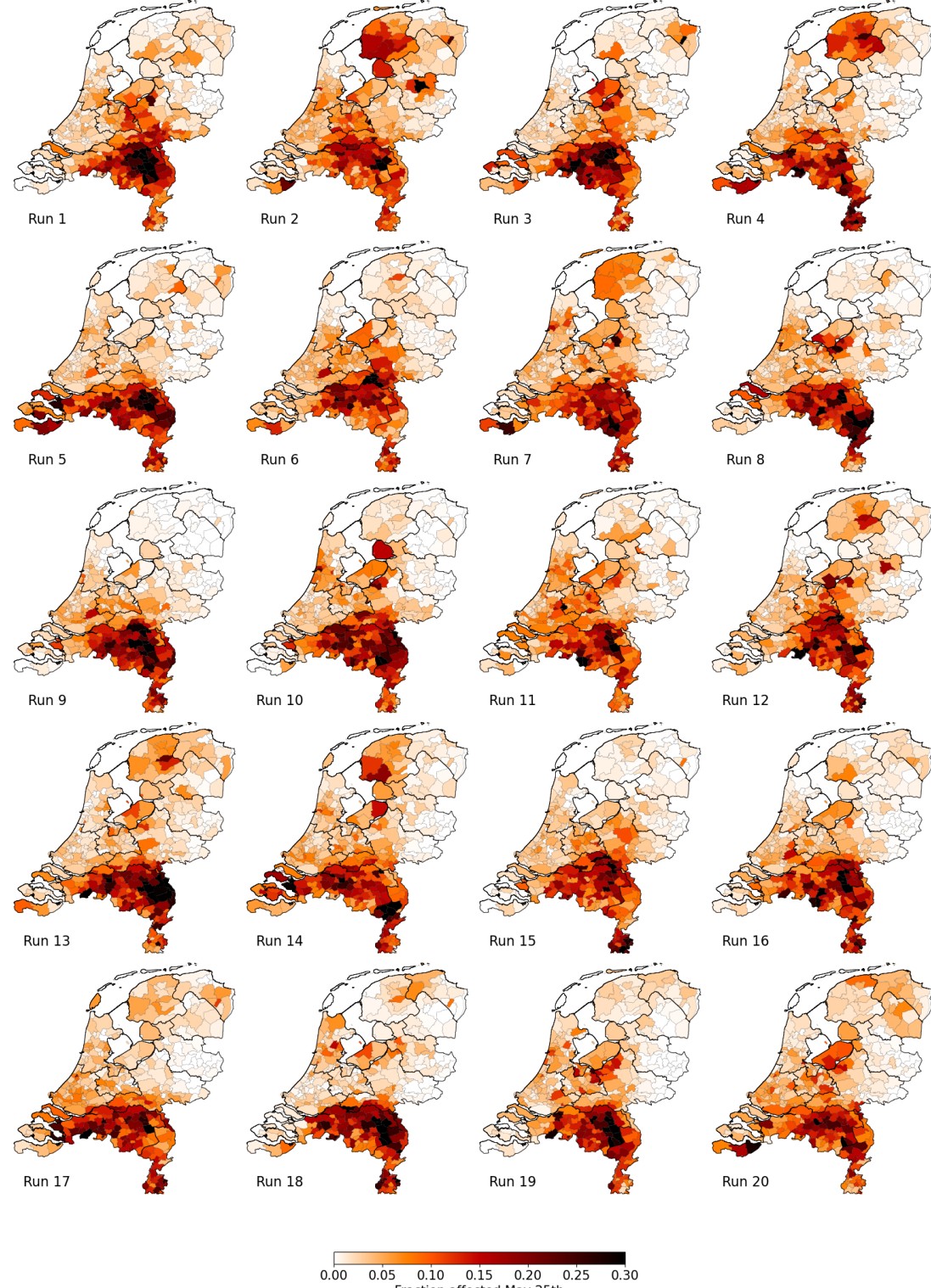

**Appendix 2—figure 3.** Percentage of affected municipality population (i.e. infectious or recovered) on May 25 using the Reference scenario, for 20 unique runs (out of an ensemble of 40). Each column represents identical mobility seeds.

## Appendix 3

### 3. Parametrization and uncertainty

#### 3.1 Sensitivity to population scale

While we have demographic and mobility information at the individual person scale, the mobility data has to be aggregated in order to simulate mobility changes and to account (in a mean-field manner) for uncertainties regarding projecting mobility information from a year earlier to a pandemic situation. In other words, with the data available, we cannot work on an agent to individual-person resolution. In addition, we need to balance with computational speed — currently, a single run with subnational interventions takes approximately 20 hours of local CPU time. Hence, we decided to work on a 1:100 level. To show the impact of this choice, with respect to, for example, a 1:75 resolution and a 1:500 resolution, we show these respective resolutions in *Appendix 3—figure 1*. In the 1:500 case, the resolution is too low, making the distribution of people at initialization and throughout the infection-transmission process too sparse — yielding a lower infection spread for similar parametrization of the $\beta$ parameters and initialization. However, when having reached sufficient population resolution (both 1:100 and 1:75), the results equilibrate onto similar levels and do not differ in terms of the overall cumulative hospital admissions (left), as well as in the spatial spread (right).

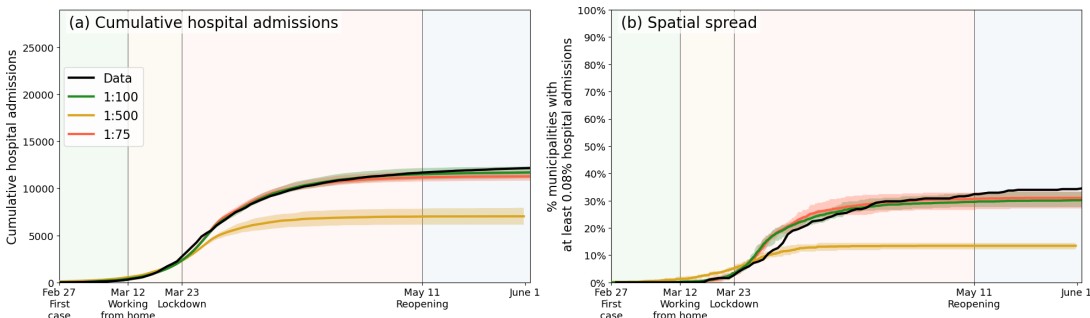

**Appendix 3—figure 1.** Same as Fig. *Figure 3* of the main text, but only including a comparison between the reference population resolution (1:100, green), an increased population resolution (1:75, red) and a reduced population resolution (1:500, orange).

#### 3.2 The mobility data and its limitations

The mobility data is collected by commercial data provider Mezuro (https://www.mezuro.com/). Importantly, we note that the data was not collected specifically for this research. The regular clients of Mezuro are municipalities that aim to have more insight in the local tourism. Mezuro infers, at any moment in time, the position of mobile phone users by determining signal towers they are closest to. The accuracy of their methodology on a ZIP code basis has been compared and validated with GPS data, meaning that the quality of the data on a municipality level is clearly sufficient. The mobile phones tracked in this research belong to the 'Vodaphone' network, and by upscaling, Mezuro infers numbers for the full country. The data comprises 2 weeks in 2019 (March 1 - March 14). Unfortunately, we neither own the data nor have the ability to collect further data, thus cannot expand the dataset beyond these two weeks.

By means of unique (anonymous) identifiers, Mezuro was able to quantify movements from one region (ZIP code or municipality) to another. In particular, the data collectors infer different types of visitors (frequent, regular and incidental) based on the movement patterns. Total numbers of moving from one municipality to another at an hourly resolution is what is provided to us. In other words, the data comprises three $M$ by $M$ origin-destination matrices (frequent, regular and incidental), where $M$ is the total number of Dutch municipalities. The 'origin' here is the home municipality, which is inferred from the statistics of the mobile phone users in a much longer time span. The values range from 34 to 54,265 total visitors in a given hour from a given municipality to a given other municipality.

While the mobility data contains valuable information, its application in our model framework clearly involves a number of uncertainties. First, the aforementioned limited time span of the database (2 weeks) and the fact that it contains data of 2019 (instead of 2020, which is our focus) require the assumption that the mobility patterns found in the 2019 data are also applicable in

2020. Lacking another data source, this assumption seems reasonable under the condition that can suitably apply scaling to the data and ex-post limit mobility in certain subnational areas to account for changes that scale linearly with regular ('2019') mobility patterns. We do not have data on changes in mobility stratified by demography or other characteristics and therefore are not able to account for this (apart from parents staying at home for their children, see Appendix 1.7). Second, the data contains aggregate movements at the municipality level and we had to infer how that translates to movements of individual agents - where, for epidemiological purposes, sequences of movements are notably important. This cannot be resolved explicitly with the data at hand and we had to rely on our calibration on this point. Third, all mobility and transmission in this model is domestic. International travel and transmission is not accounted for. We note that international travel was severely limited in the first COVID-19 wave in the Netherlands and neighboring countries, but realize that this is another source of uncertainty.

