## [Editor Report]

This useful study, based on simulations from an agent-based model of SARS-CoV-2 transmission in the first wave in the Netherlands, provides solid evidence that a subnational implementation of non-pharmaceutical interventions, with control measures varying in response to local variations in infection prevalence, would have the potential to achieve similar levels of epidemiological control to a nationally homogeneous response while reducing negative societal impacts. The work will be of interest to communicable disease epidemiologists and those involved in policy responses to epidemics.

---

## [Decision Letter]

**Decision letter after peer review:**

Thank you for submitting your article "Reducing societal impacts of SARS-CoV-2 interventions through subnational implementation" for consideration by *eLife*. Your article has been reviewed by 3 peer reviewers, including Ben S Cooper as Reviewing Editor and Reviewer #1, and the evaluation has been overseen by Neil Ferguson as the Senior Editor.

Essential revisions:

1. Sensitivity analysis to explore the impact of the 1:100 population scale assumption is needed along with clarification of why and how this approach was used.

2. Further descriptive analysis of the mobility data set (possibly as supplementary info) should be added and the authors should illustrate how that drives the local diffusion process. In particular, we strongly encourage expanding the use of the Mezuro data set beyond the short period in March 2019 to estimate effects of lockdowns on travel patterns. Further explanation of the analysis presented and categorisation into frequent, regular and incidental is also needed.

3. Additional analysis is needed to explore the consequences of decoupling the introduction of individual interventions; for example, prioritising keeping schools open longer, even as behavioural changes are introduced.

4. Additional sensitivity analyses are needed to include consideration of the impact of (i) imperfect surveillance (eg, due to asymptomatic transmission, reporting delays, etc); (ii) non-compliance, which could potentially differ for subnational versus national interventions; (iii) pathogens/variants with transmission/severity characteristics different from the original SARS-CoV-2 strain; (iv) different assumptions about the introduction of the pathogen (currently the text says it was introduced only in the south).

5. Check that numbers reported in the text and Figure 4 are in agreement.

6. Use consistent definitions throughout eg for "affected people".

7. Expand the discussion of local lockdowns to reflect experience elsewhere e.g. in Northern Italy and China, as well as a wider consideration of practical and ethical concerns including potential unintended consequences and feasibility of messaging.

8. Revise figure 2c so it is possible to distinguish the lines. The caption and legend for figure 2 are also inconsistent.

9. There are also several places where the SI needs attention either to improve clarity or to correct errors.

*Reviewer #2 (Recommendations for the authors):*

I would suggest to provide a much more detailed statistical analysis of the Mezuro data set and how this can explain the early dynamics observed in the Netherlands, with less focus on effectiveness of local lockdowns as a possible intervention.

*Reviewer #3 (Recommendations for the authors):*

Analysis:

As noted above, there are several key questions that I think warrant further analyses to increase the general relevance of the work (expanding on points raised in the Public Review):

1. The study includes three intervention types (school closure, behavioural change, and mobility restrictions) which are implemented as a package. Given the varied impact of these measures on epidemiological outcomes, it would be interesting to explore the consequences of decoupling the introduction of individual interventions; for example, prioritising keeping schools open longer, even as behavioural changes are introduced.

2. At the time of the first wave, the extent of asymptomatic transmission of SARS-CoV-2 was not well understood. Surveillance systems (in many countries) took time to establish and scale up in response to the first wave of COVID with implications for timely reporting of case numbers. This work does not currently consider the impact of imperfect surveillance on the effectiveness of subnational interventions.

3. There is an implicit (but I would consider strong) assumption that subnational interventions in a location will be as effective as national lockdown. This work does not currently consider that populations subject to localised lockdowns may be less compliant, and the extent to which this may decrease the impact of subnational lockdowns. While there is obviously limited data in the context of the Netherlands to inform such analyses, subnational interventions were deployed in other countries, such as the United Kingdom and Australia, over the last 2.5 years, providing some opportunity to observe how such interventions were managed, along with successes and challenges associated with such approaches.

4. As it stands, the scenarios demonstrate the potential effectiveness of an approach that could have been taken in response to the first wave of SARS-CoV-2 in the Netherlands. The relevance of the paper would be greatly increased by expanding the analyses to explore how effectiveness of subnational interventions may vary in the context of a different pathogen/ variant (ie, with respect to transmissibility, severity, asymptomatic infection, etc).

Discussion:

While I think there is considerable opportunity to use the model described in this manuscript to also explore the other issues mentioned above (discussion of unintended consequences, etc), I recognise that these may be considered beyond the scope of a single paper. I do feel that the discussion would be much richer if it considered some of these issues, however.

The parenthetical comment "(This would of course require local governments to be mandated appropriately and that local populations adhere to local measures)" suggests the authors are aware of issues associated with implementation issues associated with subnational interventions; however, I think a much richer discussion of this is warranted.

---

## [Author Response]

Essential revisions:1. Sensitivity analysis to explore the impact of the 1:100 population scale assumption is needed along with clarification of why and how this approach was used.

The population scale is chosen based on a trade-off between computation time and resemblance of reality in terms of specifically two factors: the mobility flows between municipalities and demographic distribution of municipalities. While the demographic distribution of municipalities is known perfectly (i.e., up to the single person scale), the uncertainties in the mobility part limit the performance benefits of increasing resolution much further than 1:100. A notable uncertainty stems from only having data on mobility flows for two weeks in 2019: the data is merely used as an indication of the general flows. Still, we agree that this parameter should be discussed more elaborately in the manuscript and that the sensitivity of the model to population scale should be pointed out.

– Added a section 3.1 in the Supplementary Material on sensitivity to population scale

– This section includes a figure (reproduced below) containing new runs with an increased population ratio of 1:75 instead of 1:100

2. Further descriptive analysis of the mobility data set (possibly as supplementary info) should be added and the authors should illustrate how that drives the local diffusion process. In particular, we strongly encourage expanding the use of the Mezuro data set beyond the short period in March 2019 to estimate effects of lockdowns on travel patterns. Further explanation of the analysis presented and categorisation into frequent, regular and incidental is also needed.

We fully agree on this point and realize that we have been too brief on the Mezuro data. We have added a section in the supplementary information describing the data in more detail. Unfortunately, we cannot expand the database beyond the short period already used: we were not part of the data collection and we currently do not have a contract with the company. Already at the time of working on this project we had a deal with a lot of privacy-related issues, even with the use of this short-period data. We do agree that there are certain uncertainties associated with having only two weeks of mobility data – these uncertainties are now addressed in the new section about this dataset.

– Added a section 3.2 specifically addressing the mobility dataset containing the details of how the data was obtained, how the data is structured, key uncertainties in the data and why we cannot expand our analysis on mobility data beyond the period in March 2019.

– The section also contains information on the categorization into frequent, regular and incidental visits and how we deal with these categories in the model framework

3. Additional analysis is needed to explore the consequences of decoupling the introduction of individual interventions; for example, prioritising keeping schools open longer, even as behavioural changes are introduced.

This is indeed an interesting question in relation to policy strategy making. This is why in Figure 3 we looked at the results of the first Dutch epidemic wave in cases where certain specific national measures were omitted, which is exploring the decoupling of individual interventions as the reviewer is asking. The resulting increase in hospital admissions is an indication of the benefits (or ‘efficacy’) of a particular measure. In subnational scenarios, we also created two other scenarios where specific changes are implemented such that their effect can be assessed: fully closing municipality borders and closing schools early (see Figure S7 in the supplementary information). Note that the question of prioritising keeping schools open longer will result in hospital admissions between the green and magenta line in Figure S7: since that range is relatively small, we do not think it is very enlightening to run a ‘middle-of-the-road’ scenario in regard to school closure (in fact, behavioural changes are still introduced in this scenario). Still, we agree that we could discuss the results of these scenarios a bit further in the direction of its policy applications.

– Rewritten the final paragraph in the conclusions about the policy implications of the scenarios in which we decouple individual interventions.

4. Additional sensitivity analyses are needed to include consideration of the impact of (i) imperfect surveillance (eg, due to asymptomatic transmission, reporting delays, etc); (ii) non-compliance, which could potentially differ for subnational versus national interventions; (iii) pathogens/variants with transmission/severity characteristics different from the original SARS-CoV-2 strain; (iv) different assumptions about the introduction of the pathogen (currently the text says it was introduced only in the south).

The reviewers make fair remarks on these sensitivities and we have added a new section 3 (supplementary information) where we elaborately discuss uncertainty and sensitivity versus both a few parameters as well as uncertainties generally associated with these type of modelling. In a pointby-point manner:

(i) Imperfect surveillance is addressed implicitly in the model. The model is calibrated to hospital admission data and in particular the time it takes between the various phases of the disease per individual: exposure, infection, recovery and also hospital admission. As these numbers are obtained from actual data they already contain such imperfections.

(ii) Non-compliance is measured within the behavioural parameter of the model (β, see Tab. 1), which is calibrated and varied across the first-wave as well. In subnational scenarios, behaviour is varied subnationally, as well: those municipalities that enter a wave-phase earlier than others, have a different β than the other municipalities.

(iii) For other pathogen and variants, with accompanying different transition and severity characteristics, both the β, the time lags (Weibull distributions, see *de Vlas, J. and L. E. Coffeng, Scientific Reports 11, 4445 (2021)*), as well as a number of severity parameters (see Tab. S2) will change. Note that only the β is a calibration parameter: other parameters can be directly deduced from clinical research.

(iv) The possibility space for the question of a different local introduction is of course infinite, but by means of illustration, as we agree that this is an important discussion, we devised an additional scenario in which the disease is introduced in Amsterdam rather than in the south of the country. This is a much more connected and central position, making the spread much faster paced.

On a general note, the results itself are fully calibrated to the case of the first epidemic wave of COVID19 in the Netherlands and will have to be re-calibrated for diseases.

– We emphasize more how these results can be used in a broader context, and the limits of the paper’s implications in the Discussion section.

– We add a section 3 in the supplementary information discussing various parameter sensitivities and structural uncertainties.

– We added a scenario in Appendix 2.3 where the disease starts in Amsterdam rather than in the south (which it was in reality). This is also shown in Author response image 1 (in brown).

**Author response image 1. sa2fig1:** 

5. Check that numbers reported in the text and Figure 4 are in agreement.

This confusion was partially caused by different quantities referred to: either number of municipalities without interventions or the number of person-days spent without interventions. The text is changed and more coherent now. A few textual errors also have been removed.

6. Use consistent definitions throughout eg for "affected people".

Affected people” clearly defined in main text now (end of section “Analysis framework”), and consistently used. It is also mentioned in the caption of Figure 2.

7. Expand the discussion of local lockdowns to reflect experience elsewhere e.g. in Northern Italy and China, as well as a wider consideration of practical and ethical concerns including potential unintended consequences and feasibility of messaging.

The specific local situation of other countries in terms of demography, mobility and in particular the evolution of the pandemic, ensures that direct copying of these results is not possible. However, we can elaborate on what we can learn from the application to the Dutch situation, in particular in relation to the new scenario runs we did where the disease was initialized in Amsterdam (see point 4iv).

– We have added discussions on situations outside of the Netherlands in the conclusions section.

8. Revise figure 2c so it is possible to distinguish the lines. The caption and legend for figure 2 are also inconsistent.

– For visualization purposes, Figure 2c is now changed into a barchart, showing the % affected at the moment of lockdown (March 23). This figure can be read much more easily.

– Caption and legend in Figure 2 has been made consistent.

9. There are also several places where the SI needs attention either to improve clarity or to correct errors.

Apart from the major changes in the Appendices on the previous points, a few other instances have been improved.

Reviewer #2 (Recommendations for the authors):I would suggest to provide a much more detailed statistical analysis of the Mezuro data set and how this can explain the early dynamics observed in the Netherlands, with less focus on effectiveness of local lockdowns as a possible intervention.

As we replied above, we agree that more discussion is needed on statistical robustness and sensitivity.

This has been added.

Reviewer #3 (Recommendations for the authors):Analysis:As noted above, there are several key questions that I think warrant further analyses to increase the general relevance of the work (expanding on points raised in the Public Review):1. The study includes three intervention types (school closure, behavioural change, and mobility restrictions) which are implemented as a package. Given the varied impact of these measures on epidemiological outcomes, it would be interesting to explore the consequences of decoupling the introduction of individual interventions; for example, prioritising keeping schools open longer, even as behavioural changes are introduced.

See point [3] of “Essential Revisions” above.

2. At the time of the first wave, the extent of asymptomatic transmission of SARS-CoV-2 was not well understood. Surveillance systems (in many countries) took time to establish and scale up in response to the first wave of COVID with implications for timely reporting of case numbers. This work does not currently consider the impact of imperfect surveillance on the effectiveness of subnational interventions.

See point [4] of “Essential Revisions” above.

3. There is an implicit (but I would consider strong) assumption that subnational interventions in a location will be as effective as national lockdown. This work does not currently consider that populations subject to localised lockdowns may be less compliant, and the extent to which this may decrease the impact of subnational lockdowns. While there is obviously limited data in the context of the Netherlands to inform such analyses, subnational interventions were deployed in other countries, such as the United Kingdom and Australia, over the last 2.5 years, providing some opportunity to observe how such interventions were managed, along with successes and challenges associated with such approaches.

See point [4] of “Essential Revisions” above.

4. As it stands, the scenarios demonstrate the potential effectiveness of an approach that could have been taken in response to the first wave of SARS-CoV-2 in the Netherlands. The relevance of the paper would be greatly increased by expanding the analyses to explore how effectiveness of subnational interventions may vary in the context of a different pathogen/ variant (ie, with respect to transmissibility, severity, asymptomatic infection, etc).

See point [4] of “Essential Revisions” above.

Discussion:While I think there is considerable opportunity to use the model described in this manuscript to also explore the other issues mentioned above (discussion of unintended consequences, etc), I recognise that these may be considered beyond the scope of a single paper. I do feel that the discussion would be much richer if it considered some of these issues, however.The parenthetical comment "(This would of course require local governments to be mandated appropriately and that local populations adhere to local measures)" suggests the authors are aware of issues associated with implementation issues associated with subnational interventions; however, I think a much richer discussion of this is warranted.

The reviewer is correct and highlights a few important limits: comparability to other countries, other variants, mandates and compliance. Although to a certain degree, the latter can be assumed to be comparable between national and regional policies (we are still talking about the first COVID-19 wave in the Netherlands), indeed realizing that high regional differences may lead to frustration, the application of this analysis to other countries with own geographies, initialization, variants and institutions is difficult to assess. As part of point [4] of “Essential Revisions” above, we therefore aim to extend the discussion on these points further – albeit qualitatively. The quality of the model, being already rather complex, relies on the calibration to high resolution data on mobility, demography and hospital admissions. Thus, even though the framework could be reproduced in the future by means of similar calibration and data of another country or situation, from the current state we cannot make quantitative statements of such other situations.